# Critical slowing down as a biomarker for seizure susceptibility

Matias I. Maturana [1,2 ✉], Christian Meisel[3,4], Katrina Dell[1], Philippa J. Karoly[5], Wendyl D'Souza [1], David B. Grayden [6], Anthony N. Burkitt [6], Premysl Jiruska [7,8], Jan Kudlacek[7,8,9], Jaroslav Hlinka [10,11], Mark J. Cook [1,5], Levin Kuhlmann [1,12,13] & Dean R. Freestone[2]

The human brain has the capacity to rapidly change state, and in epilepsy these state changes can be catastrophic, resulting in loss of consciousness, injury and even death. Theoretical interpretations considering the brain as a dynamical system suggest that prior to a seizure, recorded brain signals may exhibit critical slowing down, a warning signal preceding many critical transitions in dynamical systems. Using long-term intracranial electro-encephalography (iEEG) recordings from fourteen patients with focal epilepsy, we monitored key signatures of critical slowing down prior to seizures. The metrics used to detect critical slowing down fluctuated over temporally long scales (hours to days), longer than would be detectable in standard clinical evaluation settings. Seizure risk was associated with a combination of these signals together with epileptiform discharges. These results provide strong validation of theoretical models and demonstrate that critical slowing down is a reliable indicator that could be used in seizure forecasting algorithms.

[1] Department of Medicine, St Vincent's Hospital, The University of Melbourne, Melbourne, Australia. [2] Seer Medical, Melbourne, Australia. [3] Department of Neurology, University Clinic Carl Gustav Carus, Dresden, Germany. [4] Boston Children's Hospital, Boston, MA, USA. [5] Graeme Clark Institute, The University of Melbourne, Melbourne, Australia. [6] Department of Biomedical Engineering, The University of Melbourne, Melbourne, Australia. [7] Department of Physiology, Second Faculty of Medicine, Charles University, Prague, Czech Republic. [8] Department of Developmental Epileptology, Institute of Physiology, Czech Academy of Sciences, Prague, Czech Republic. [9] Department of Circuit Theory, Faculty of Electrical Engineering, Czech Technical University in Prague, Prague, Czech Republic. [10] Institute of Computer Science of the Czech Academy of Sciences, Prague, Czech Republic. [11] National Institute of Mental Health, Klecany, Czech Republic. [12] Faculty of Information Technology, Monash University, Clayton, Victoria, Australia. [13] Centre for Human Psychopharmacology, Swinburne University of Technology, Hawthorn, Victoria, Australia. ✉email: matiasim@unimelb.edu.au

The unexpected nature of epileptic seizures represents the major clinical disability of epilepsy[1]. The mechanisms underlying the transition from a normal to a seizure state are currently an open question[2–4]. Unraveling the mechanisms underlying seizure generation could form the basis of much needed new treatment strategies, particularly for patients where existing treatments are ineffective.

Abrupt state changes in natural systems, including the onset of seizures, can, in principle, be due to critical transitions[5]. A characteristic of a system that is approaching a critical transition is a phenomenon called "critical slowing down". Critical slowing down refers to the tendency of a system to take longer to return to equilibrium after perturbations, indicated by an increase in signal variance and autocorrelation. Generally, critical slowing down can be expected if a system is driven towards the transition point at a moderate pace[6] and if the basin of attraction around the equilibrium point can be approximated by linear-stability analysis[7]. It has been observed in many systems, including cell population collapse in bacterial cultures[8] and crashes in financial markets[9]. Critical transitions have been employed to describe neural systems, such as onset of depression[10], pharmacologically induced cortical state changes[11–13], onset of spiking in neurons[14], and termination of epileptic seizures[15].

It has been hypothesized that the rapid transition from normal brain activity to an epileptic seizure also corresponds to a critical transition[3,4,16–19]. However, empirical evidence for this hypothesis has been missing in humans, which may be due to lack of long-term recordings as well as intra-patient and inter-patient variability. Empirical validation of critical slowing down in humans would provide vital support for current theoretical models of seizure generation and of the dynamics of the brain in general. Furthermore, it could aid in forecasting seizures and potential titration of epilepsy therapies.

Computational neural models are powerful tools for studying the dynamics of the brain. Numerous computational models of epilepsy suggest that seizures reflect a change in brain state via a critical transition[16,18,20]. Mathematical analyses of dynamic systems, combined with simulations, enable classification of bifurcations and critical transitions[21]. Simulations enable controlled experiments that vary the parameters of the model and reveal statistical markers that are representative of transition susceptibility, such as increases in signal variance and autocorrelation. While some methods have been developed to track control parameters (variables that drive changes in state) from clinically captured electroencephalography (EEG) in epilepsy[22,23], this approach is not straightforward. Alternatively, tracking the statistical markers related to critical slowing down in clinical EEG recordings may constitute a direct test of the hypothesis that seizures occur via a critical transition.

In this paper, we test the hypothesis that markers of critical slowing down can be used as a biomarker of seizure susceptibility. We examine hallmark signals of critical slowing down using a continuous intracranial electroencephalography (iEEG) dataset from the first-in-human trial of an implanted seizure prediction device that was recorded over multiple years[24]. As the markers of critical slowing down can potentially change over very long timescales, the long duration dataset used for this analysis provides a unique opportunity where critical slowing down in humans can be robustly investigated. We show that the autocorrelation and variance of the iEEG signals are modulated by patient-specific cycles over long temporal scales. Furthermore, we show that modulations of the variance and autocorrelation are related to seizure susceptibility—a probabilistic propensity to have seizures.

## Results

**Conceptualization of critical slowing down in epilepsy.** In epilepsy, seizure events could be described as a "phase" or "critical" transition, based on deterministic dynamics, where the brain shifts from a normal to a seizure state[5]. Assuming that the system dynamics are driven towards the transition point at a moderate pace[6] and that the basin of attraction around the stable region can be approximated by linear-stability analysis[7], approaching the critical transition is expected to be accompanied by increases in signal variance and autocorrelation, i.e. the signatures of critical slowing down[5]. Although there are different model specifics and possible paths leading to a state change[18,21], it may thus be possible to detect the occurrence of a critical slowing down close to the seizure onset[3]. We describe here a model to demonstrate how transitions may occur with the view that this concept may hold for a wider class of models.

Several models of epilepsy describe the change in brain state from normal to seizure as a bifurcation that occurs as the system crosses a critical point[16–19,21,25]. Figure 1a illustrates a one-dimensional nonlinear dynamical system, where the state $z$ is modulated by the driving parameter $k$. We can think of $z$ as being a fast-changing property of the iEEG signal; for example, it could represent the time-varying mean membrane potential of pyramidal cells averaged locally in space. The parameter $k$ represents the driving element, which could represent the response of the brain to a variety of factors such as medication, sleep, or metabolic processes. The lines in Fig. 1a (colored and black dashed) represent the fixed-points or equilibrium values taken by $z$ for any given value of $k$. The Hartman–Grobman theorem[26] suggests that close to a fixed-point, the system's dynamics can be reduced to a simpler linearized system. The color of the lines represents the time constants associated with the linearized system, and therefore describe the response dynamics of the system close to the equilibria.

Given this system, there are two possible ways to transition from the s1 (normal) to s3 (seizure) states. The first involves varying the driving parameter $k$ positively such that the system approaches and passes s2 (orange dashed arrow). In this case, we should observe critical slowing down—a slowing of the signals monitored, which is characterized by an increase in autocorrelation and variance (see Supplementary Note 1 for details). The occurrence of critical slowing down has been shown to occur under the assumption of moderate noise; noise that is too large can cause a transition to the new state[6]. The second involves a perturbation (e.g. noise) that kicks the system across the unstable threshold (dashed black line) and into the seizure state, s3 (green dashed arrow). In this case, a state transition still occurs but critical slowing down may not be expected due to the rapid push into a new state.

The clinical definition of a seizure onset is often subjective. In our example, a shift into state s3 is an unequivocal onset, but clinicians also recognize the *earliest electrographic change* which may precede clinical symptoms by seconds to minutes[27]. The electrographic change is characterized by an evolution of the iEEG signal in time and frequency, which could be described by changes observed through critical slowing down. Therefore in our example, the seizure onset begins at s1 and is followed by a transition regardless of the path taken to s3.

Assuming the linear approximation captures the basin of attraction around the stable region, then the system's response function will be directly proportional to the autocorrelation function of the signal, from which the time constant can be estimated (see Supplementary Note 1 for proof). Thus, analyzing the system with regards to the autocorrelation function allows us to predict how the signals should evolve when a seizure occurs (Fig. 1b). In the first case, we should observe an increase in the

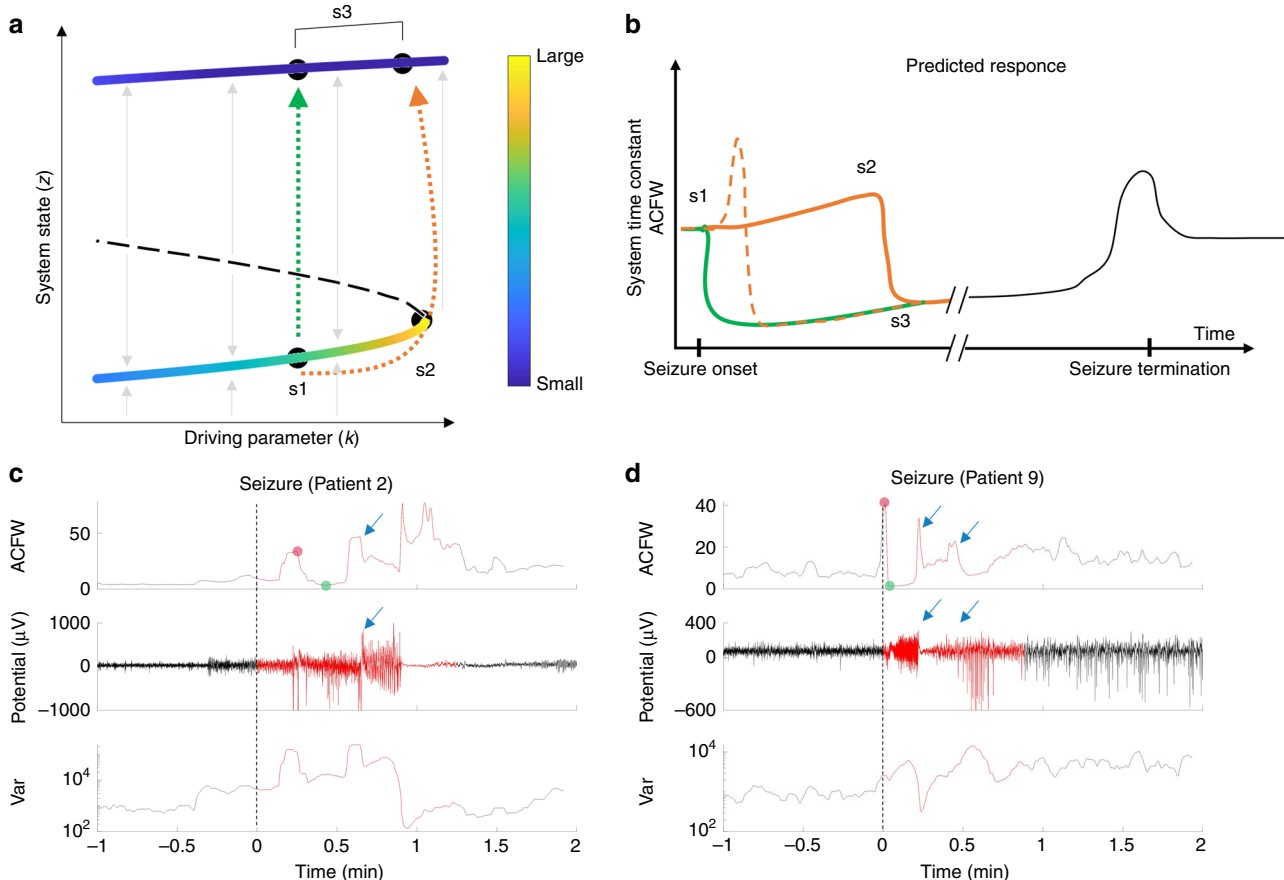

**Fig. 1 Conceptualization of critical slowing down with regards to intracranial EEG (iEEG) signals. a** A bifurcation diagram showing the system's steady states. The seizure state is assumed to lie along the top line, at points s3. The state *z* represents a property of the iEEG signal, which could be the mean action potential firing rate or mean membrane potential of pyramidal cells (believed to be the principle generators of the iEEG signal). Two stable steady states (colored lines) are separated by an unstable steady state (black dashed line). The color represents the linearized system's time constant at a given fixed point. The time-constant is highest when the system is close to the critical point, s2, and is smaller away from the critical point. Starting at point s1, two possible seizure onset routes are shown (green and orange dashed arrow). **b** By monitoring the system time constant, or the signal autocorrelation (ACFW), we expect to observe a consistent profile for both types of state transitions. In the perturbation-mediated transition (green), we expect to see a sharp decrease in autocorrelation close to the seizure onset. In the transition that occurs via the critical point, we expect to see a gradual (solid orange) or fast (dashed orange) increase in the autocorrelation amplitude depending on the speed of approach to the critical point, followed by a sharp drop. **c** An example seizure from Patient 2 (middle plot) where a seizure likely occurs via a slow approach to the critical transition, similar to the solid orange curve in **b**. **d** An example seizure from Patient 9 where a seizure likely occurs via a fast approach to the critical transition, similar to the dashed orange curve in **b**. Dashed lines in **c** and **d** represent the clinically marked seizure onset time. Blue arrows represent other likely transitions that occur during the seizure.

width of the autocorrelation function in the period leading up to s2. The speed at which the transition is approached will characterize the shape of the autocorrelation prior to the transition (orange dashed and solid curves). In many cases, the transition will be followed by a sharp drop in the width of the autocorrelation function assuming that the seizure state is characterized by a smaller time constant. In the second case, critical slowing down may not observed due to a rapid perturbation that causes a transition. Assuming that s1 is relatively close to s2, then in this case we should only observe a sharp drop in autocorrelation function width close to the seizure onset (green curve). The seizure termination has also been found to follow a critical transition[15], hence it is possible that the seizure termination also follows a similar pattern (black line).

**Seizure onset as a critical transition at short time scales.** A sample seizure from a patient showing a gradual increase in the signal markers, in line with a slow approach towards the critical point is shown in Fig. 1c. This is in agreement to trajectory one,

where the seizure onset begins at s1, gradually passes through s2, and transitions to s3. The transition point is also characterized by a clear change in the electrographic activity from low amplitude fast activity, to large amplitude spiking. A sample seizure showing a fast approach to the critical point is shown in Fig. 1d. In this case the seizure exhibits a sharp increase in autocorrelation, followed by a sharp drop at seizure onset.

These examples highlight the high dimensional nature of the system, where transitions may occur even within the seizure itself (blue arrows). These transitions are accompanied by obvious changes in the electrographic activity during the seizures. Transitions within the seizure suggests that s3 has other dimensions (imagine a dimension into the page in Fig. 1a) that may also contain critical points.

The presence of critical transitions were analyzed for all seizures in each patient (Supplementary Fig. 2). We observed that 13 of the 14 patients demonstrated the characteristic changes close to the seizure onset as described by a critical transition. Most patients had seizures with a fast onset transition similar to

the example in Fig. 1d (see arrows in Supplementary Fig. 2). Two patients had a slow transition into seizure similar to the example in Fig. 1c (Patients 2 and 4). Three patients had little signs of critical slowing down prior to seizure, instead showing a sharp decrease similar to the perturbation-mediated seizure onset (Fig. 1a and b, green; Patients 5, 7). Lastly, one patient with very few clinical seizures demonstrated no signs of a critical transition at the seizure onset (Patient 12).

The patient average in Supplementary Fig. 2 compares the peak and subsequent trough in the autocorrelation function width (ACFW) during seizures, to a baseline period 5 min prior to the seizure (e.g. red and green dots in Fig. 1c, d). Across all seizures in all patients (excluding Patient 12), the peak in ACFW was significantly higher than baseline, and the subsequent trough was significantly lower.

These examples provide strong evidence of critical transitions close to the seizure onset. Close to the transition, we expect to see these effects since the seizure dominates the system's dynamics. However, on longer time scales and during periods far from seizures, it is possible that other effects (like sleep) dominate the brain's dynamical behavior. Recent observations suggest that seizure propensity are modulated over long timescales[28,29]. As a potential measure of seizure propensity, we next asked if and how the measures of critical slowing down were related to those long cycles and timescales, and whether monitoring metrics of critical slowing down could be used to forecast seizures.

**Signatures of critical slowing down on long time scales.** Continuous iEEG recordings obtained from a clinical trial of a seizure prediction device from 14 patients were used in this study[24]. The device comprised of an array of 16 electrodes that were placed on the surface of the brain near the presumed epileptogenic zone

(Fig. 2a). After pre-processing, a total of 2871 seizures were analyzed (Table 1).

We sought to investigate the relationship between critical slowing, epileptiform spikes (a known biomarker of epilepsy) and seizures. We hypothesized that the likelihood of seizures would be related to the modulation of the autocorrelation and the variance signals. Furthermore, this modulation may also be linked to

**Table 1 Patient data summary.**

| Patient | Total seizures | Mean seizure rate (seizures/ day) | Number of days | Data dropout (%) |
|---|---|---|---|---|
| 1 | 151 | 0.20 | 767 | 35.1 |
| 2 | 32 | 0.04 | 730 | 19.6 |
| 3[a] | 374 | 0.70 | 557 | 78.3 |
| 4 | 22 | 0.09 | 233 | 26.2 |
| 5 | 9 | 0.03 | 273 | 40.11 |
| 6 | 71 | 0.16 | 441 | 2.95 |
| 7 | 313 | 1.70 | 185 | 11.93 |
| 8 | 466 | 0.84 | 558 | 35.7 |
| 9 | 202 | 0.51 | 395 | 36.4 |
| 10 | 545 | 1.46 | 373 | 6.9 |
| 11 | 461 | 0.64 | 722 | 36.6 |
| 12 | 13 | 0.02 | 729 | 3.4 |
| 13 | 497 | 0.67 | 747 | 24.3 |
| 14 | 12 | 0.02 | 627 | 32.8 |
| 15 | 77 | 0.17 | 466 | 1.25 |
| Total[b] | 2871 | | 7246 | Mean[b] 22.37 |

The total number of seizures, mean seizure rate, the number of recorded days and the percentage of data dropouts are tabulated.
[a]Patient 3 was excluded owing to high dropouts.
[b]Totals and means exclude Patient 3.

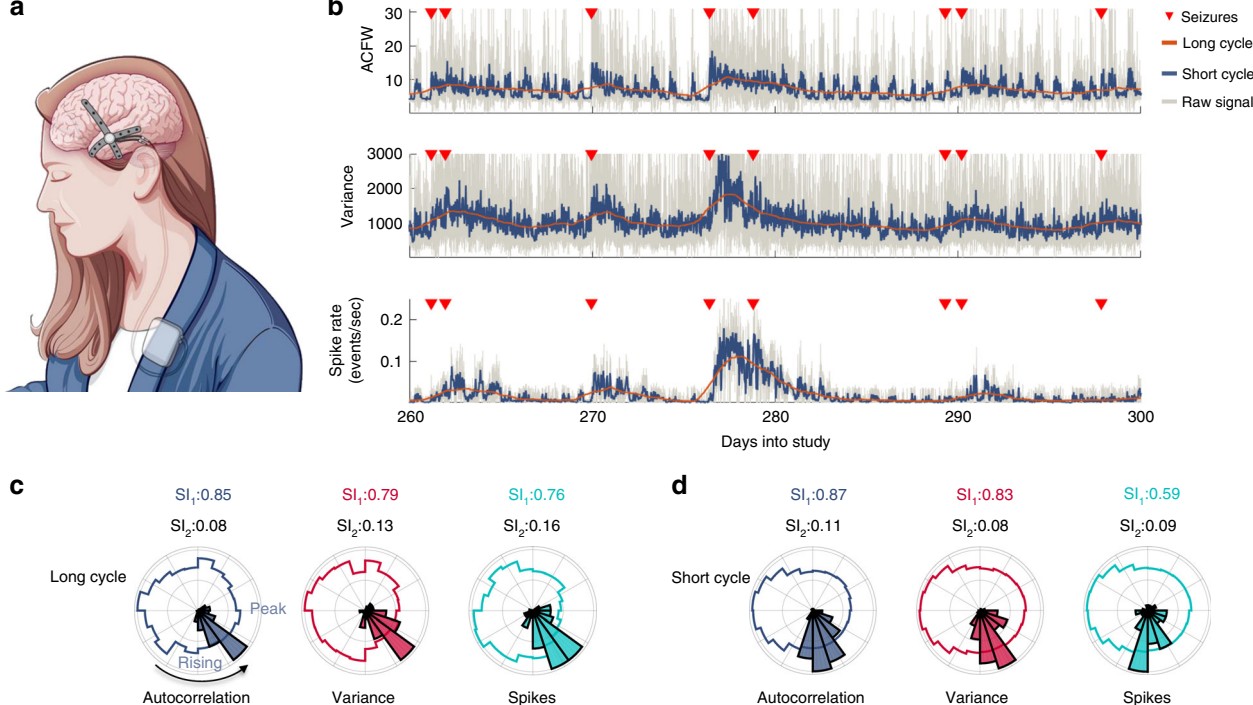

**Fig. 2 Details of analysis of Patient 1. a** An illustration of the implanted electrodes that captured continuous EEG signals from the surface of the brain at 16 different locations. **b** The autocorrelation (top), variance (middle), and spike rate (bottom) signals were filtered using a moving average filter to reveal short and long rhythms. For clarity, here we show an anti-causal filter with the short rhythm shown prior to subtraction of the long rhythm. Seizures (red triangles) preferentially occurred on the rising phases of the signals. **c** Normalized polar histograms of the seizure phases for the long cycles demonstrated that most seizures occurred on a narrow phase of the autocorrelation (gray), variance (red), and spike rate (cyan) signals. The synchronization indices for the signals (black) and histograms (colored) are shown above each plot. **d** Normalized polar histograms of the seizure phases for the short cycles.

known changes in the rates of epileptiform spikes[28,29]. Herein, we refer to *spike rates* as the rate of epileptiform spikes detected on each electrode.

It is known that seizures and epileptiform spikes are often modulated by circadian and multidien (>24 h) rhythms in a patient-specific manner[28,29]. We hypothesized that the markers of critical slowing down may also follow a rhythmic process. To assess the temporal properties of the autocorrelation signal, a Fourier transform (FT) of the autocorrelation signal was computed. The peaks in the FT demonstrated that all patients had strong circadian rhythms, and some also had strong multidien rhythms with peaks between 3 and 30 days (Supplementary Fig. 3).

Using the information derived from the FT, we divided the data into two temporal scales: short rhythms with periods of one day or less and long rhythms with periods with >2 days. We investigated the relationship between seizures and the spike rates, autocorrelations, and variances derived from the iEEG on these two temporal scales.

**A case study of critical slowing down in epilepsy**. Figure 2b shows the raw autocorrelation, variance and spike rate signals (gray), along with the long (orange), and short (dark blue) cycles of a sample channel for a representative patient, Patient 1. The coincident signal phases and seizure times were used to assess the relationships between seizures and the signals. Figure 2c illustrates the relationship between seizures and the long cycles. In each polar plot, the colored lines represent the normalized distribution of phases of the entire signal and the histogram represents the distribution of phases at the sample prior to the seizure times relative to the autocorrelation (gray), variance (red), and spike rate (cyan) signals. For this patient, seizures predominately occurred on the rising phases of the three signals and rarely occurred on the falling phases. Figure 2d similarly illustrates the relationship between seizures and the short cycles, also showing a predominance of seizures occurring on the rising phases of the signals.

Above each polar plot are the synchronization indices (SIs) for the signals (black) and for the seizure histograms (colored). The SI (not to be confused with signal synchrony across channels) is a measure of phase uniformity in a signal (the tendency of a signal to have all phases uniformly distributed on a circle), or the synchrony between a rhythmic signal and events (seizures). For the seizure histograms, a SI close to one indicates greater synchrony at a single phase. For Patient 1, the high SI across all the three signals demonstrated that there was a strong relationship between seizures and a rising phase of the signal cycles.

**Critical slowing in 14 patients**. Seizures tended to occur on the rising phase of the autocorrelation and variance signals (Fig. 3a). While the cycles observed in the three measures might not be directly linked to epilepsy (i.e. a circadian cycle would be expected irrespective of epilepsy), the results show there is a strong relationship between the phase of the cycles and seizure onset. It is possible that an increase in autocorrelation and variance is descriptive of an increase in overall brain excitability. A gradual increase in the autocorrelation and variance signals, suggestive of critical slowing down, was observed in the tens-of-minutes to hours prior to lead seizures in most patients (9 of 14 patients; Supplementary Fig. 4). In a few patients (Patients 8, 9, 11, and 13), most seizures occurred when the autocorrelation and variance signals were decreasing (Supplementary Figs. 12, 13, 15, and 17). The circadian cycle was the dominant rhythm in these patients as illustrated by a low SI for the long rhythms. In these patients, it is possible that the circadian cycle modulates whether

the brain exists in a mono-stable or bi-stable regime, thereby controlling when seizures can occur. This scenario is described in more detail in the Supplementary Note 2.

The SI was used to identify the electrodes that best captured the relationships between seizures and the underlying long and short cycles. The SIs across patients for long and short cycles are shown in Fig. 3a. The SIs were >0.5 for the short cycles for nearly all patients, suggesting a strong relationship between seizures and the short rhythms. Some patients also had a strong relationship between seizures and the long cycle (see Supplementary Figs. 6–19 for patient summaries). The average short cycle duration was 0.64 ± 0.16 days, and the average long cycle duration was 10.5 ± 4.1 days (Supplementary Fig. 20A).

Visual inspection of the autocorrelation signal in each patient showed that the signal tended to be similar across electrodes. Conversely, for the spike rate signal, the rates tended to be variable across electrodes. We quantified the similarity of the three signals by computing a mean cross-correlation for the three signals independently across all channel pairs (Fig. 3b). The autocorrelation signal was significantly higher than the variance and the spike rate signals suggesting that the autocorrelation signal was consistent across the recorded brain areas. The autocorrelation might therefore represent a measure describing a state change throughout the brain, rather than a change in a localized brain region as is the case with epileptiform spikes.

It is well recognized that seizures tend to cluster[30,31]. We sought to examine the relationship between the autocorrelation and variance signals and seizure clusters. We found that 9 of the 14 patients had clusters of seizures that occurred within a short interval of a lead seizure. For a subset of these patients, the autocorrelation and variance signals slowly tended back to baseline after lead seizures over a period (hours to days) well beyond the duration of the individual seizures. During this time, there was an increased susceptibility to more seizures. Figure 3c shows an example of this relationship for Patient 1 (see also Fig. 3d, e top row for Patients 6, 10, and 12).

For another subset of patients, both the autocorrelation and variance signals steadily increased after lead seizures. During this period there was an increased seizure susceptibility (Fig. 3d, e middle rows), which was reduced when the autocorrelation and variance signals began to decrease.

The increases in autocorrelation and variance following a lead seizure may be representative of an overall increase in brain excitability, where seizures are more likely to keep occurring. Exceptions to this trend were Patients 7 and 13. Patient 7 showed an increase in seizure rate following lead seizures, despite decreased autocorrelation and almost no change in variance. For Patient 13, there was only an increase in variance and almost no change in autocorrelation (Fig. 3d, e bottom rows).

**Seizure forecasting**. We evaluated the performance of a seizure forecasting algorithm based on the detected rhythms of the autocorrelation, variance, and spike rate signals. We consider two approaches:

- *Method M1*: Anti-causal filtering was used and potential forecasting performance was evaluated using all the available data. In this case, we evaluated the optimal level that a forecaster may perform using within-sample optimization, where prior knowledge of seizure susceptibility relative to the phase of the signals was computed using all the data. This method tests the performance of combining autocorrelation, variance, and spike rate signals in a forecaster and represents the best possible performance outcome of a forecaster.

- *Method M2*: Seizure rhythms were computed iteratively with a causal filter such that the forecaster for a given patient was

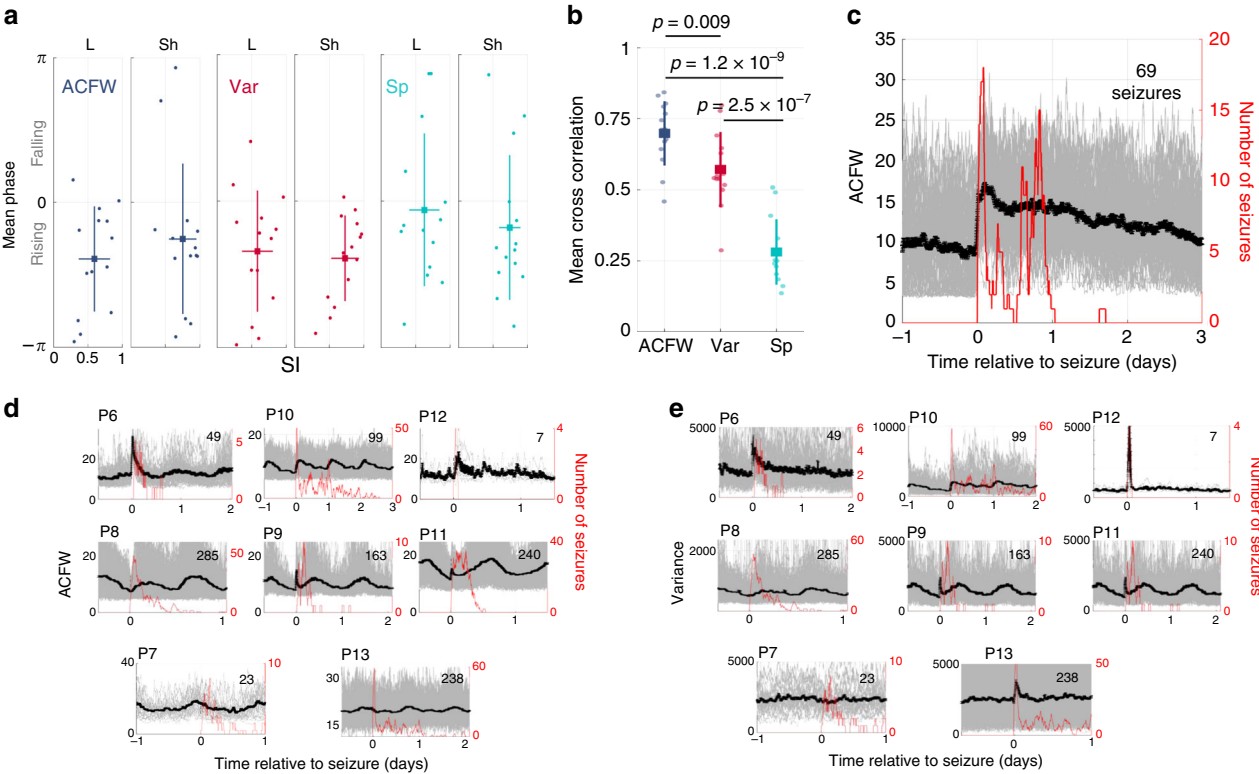

**Fig. 3 Critical slowing down and seizure clusters. a** The synchronization indices (SIs) and mean phases at the sample prior to seizures for long (L) and short (Sh) cycles. A strong relationship between the three signals and seizures were observed as depicted by the high SI values. On average, seizures occurred on the rising phase of the signals. **b** The similarity in autocorrelation (ACFW), variance (Var), and spike rate (Sp) signals across electrodes were compared by computing the mean cross correlation of each signal and comparing across patients. There was a significant effect of signal type ($F_{2,13} = 58.61$, $p = 2 \times 10^{-10}$), and a post-hoc analysis showed that the autocorrelation signal was significantly different to the variance ($p = 9 \times 10^{-3}$) and the spike rate signals ($p = 1 \times 10^{-9}$). In **a** and **b**, each dot represents a result for one patient ($N = 14$ patients); boxes indicate means across patients and lines indicate ±one standard deviation. In **b**, statistical comparisons were computed using a balanced two-way ANOVA corrected with a Tukey–Kramer multiple comparisons test. **c** Patient 1 had 69 seizure clusters. Gray lines shows the raw autocorrelation from individual seizures. **d**, **e** Autocorrelation **d** and variance **e** relative to the time from lead seizures for the remaining patients with seizure clusters. The numbers inset in each subplot denotes the number of lead seizures that occurred in clusters. In **c–e**, black lines denote the mean autocorrelation with standard error bars. Red lines denotes a moving sum of seizures occurring after the lead seizure computed over a 2 h window.

based on information provided only by previous seizures. This approach tested out-of-sample forecasting performance in a pseudoprospective manner. Forecasting using this method began after the 10th seizure. This method represents a forecaster based on the same signals but computed in a manner that is applicable to a clinical setting, where the algorithm learns iteratively as data becomes available.

The relationships between seizures and signal phases were used to calculate the probability of a seizure. Figure 4a depicts the probability of a seizure occurring for Patient 1 using Method M1. Specifically, this is the probability of a seizure given the phases of the long and short cycles

$$P(\text{Seizure}|A_1, A_2) \qquad (1)$$

where $A_1$ and $A_2$ refer to the phases of the long and short cycles, respectively. From the probability distribution, the seizure probability versus time was calculated by multiplying the individual distributions under the assumption that each probability distribution was independent, since there was insufficient data to characterize the joint distributions accurately (Fig. 4b, top). In an approach similar in nature to the original trial of the seizure advisory system that gave rise to the data considered here[24], our forecaster was designed such that, at any given time, a patient would be placed in a risk category: low, medium, or high

risk. In practice, indicating the current risk level to a patient can be used to help guide their daily activities and encourage them to move to safety when seizure risk is high.

Using the seizure probability described above, two thresholds that optimally separated the low, medium, and high-risk categories were computed and used to categorize risk state over time (Fig. 4b, bottom). A risk level defines the risk of a seizure occurring after the next sample, which provides a 2–4 min prediction horizon. For Patient 1 and using Method M1, 3% of seizures occurred during low risk, and 90% occurred during high risk. The proportion of time spent in the high-risk category was 3% and in the low-risk category was 95%.

To simulate a realistic situation that could be applied to clinical practice, we computed and updated the seizure probability distributions and risk levels iteratively after each new seizure (Method M2). Figure 4c shows the risk level assigned to each seizure (gray) and a moving average over five seizures (black) for Patient 1. The average risk level assigned to all seizures using Method M2 was 2.7. For Patient 1 and using Method M2, 15% of seizures occurred during low risk and 83% during high-risk categories. The proportion of time spent in the high-risk category was 8% and in the low-risk category was 91%. A receiver operating characteristic (ROC) was also computed for both methods, which is presented in log scales to emphasize the performance during periods of low risk (Supplementary Fig. 20B).

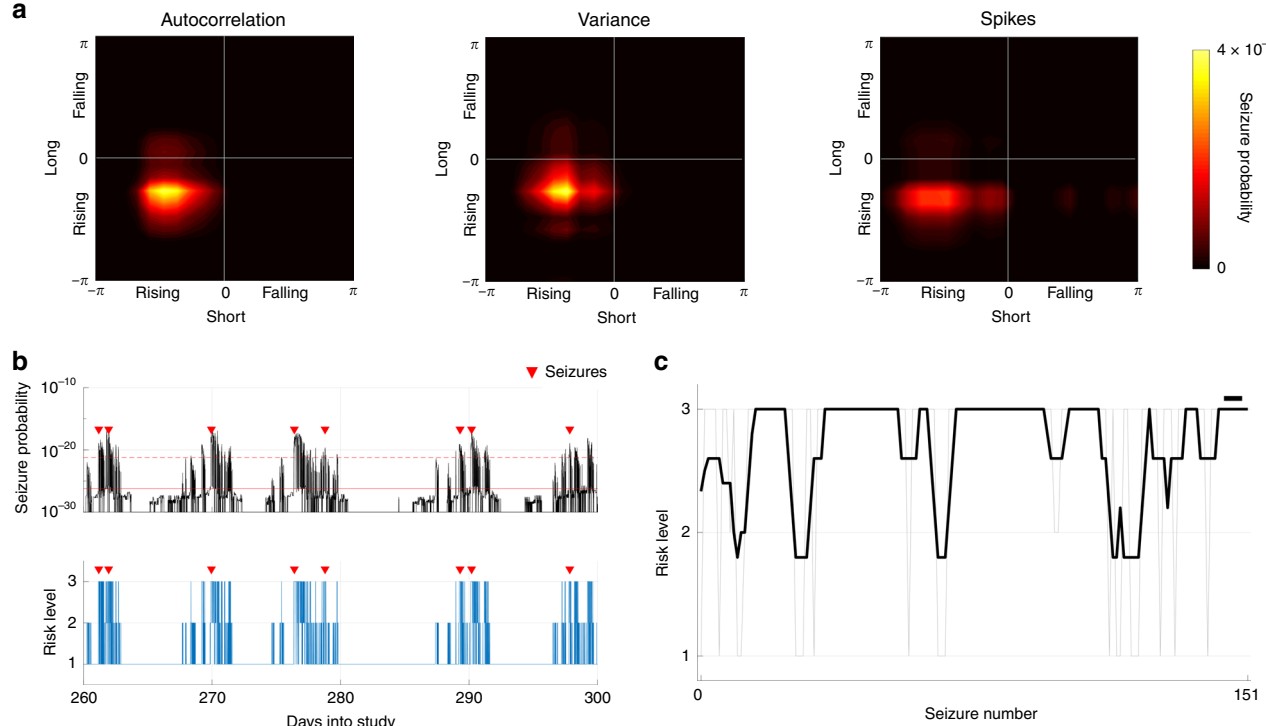

**Fig. 4 Seizure forecasting examples for Patient 1. a** The probability distributions of seizures given phases of the long and short cycles for the autocorrelation, variance, and spike rate signals. Brighter color corresponds to higher probability. **b** The distributions from **a** were used to compute the probability of a seizure over time (top). Two thresholds separated low from medium risk (solid red) and medium from high risk (dashed red). The risk levels over time are shown in the bottom plot indicated by the heights of the blue bars; seizure times are indicated by red triangles. **c** Pseudoprospective Method M2, where risk level at the time of each seizure is shown by the gray line and the black line denotes the five-seizure moving average risk (the black bar on the top-right denotes the length of the moving average).

Figure 5a illustrates Method M2 approach applied to the data of the remaining patients (except for Patient 5 due to too few seizures). The average risk level assigned to seizures was 2.5 or greater for every patient, demonstrating that Method M2 achieved good performance. Figure 5b, c quantifies the performances of the two forecasters and compares it to a random predictor. Significantly better predictions were observed using Methods M1 and M2 compared to the chance model. No significant differences in the amount of time spent in each risk state was observed.

To compare the overall performances of the methods, the product of the proportion of seizures in the high-risk category and the proportion of time spent in the low-risk category was used, which is hereafter called the "performance product". In the ideal case, the performance product would be close to one. Methods M1 and M2 performed significantly better than chance. There was no significant difference between Methods M1 and M2 (Fig. 5d). ROC curves were also computed across all patients (Supplementary Fig. 20C, D).

Seizure forecasting performance was compared using measures from critical slowing down versus spike rates alone, and then the combined measures (Supplementary Fig. 21), and to the performance of the original trial (Supplementary Table 1). Method M1 achieved a higher sensitivity and specificity overall compared to Method M2 and the original trial. Finally, we compared our pseudoprospective forecasting results to other pseudoprospective algorithms previously developed on the same dataset. This included a machine-learning algorithm[32], a predictor based on circadian rhythms and logistic regression[33], and multiple algorithms based on a crowd-sourced approach to seizure prediction[34]. Method M2 scored higher on sensitivity

(seizures in high risk), and a lower time-in-high for all patients except Patients 7 and 14 (Supplementary Table 2).

## Discussion

Prior to this study, seizures were theorized to occur via a critical transition[3,16,18,20]. However, there has been little empirical evidence for this hypothesis in humans. We previously demonstrated that critical slowing down is present prior to seizures in work focused on in vitro and in vivo models[35]. The current paper is a unifying progression where we have investigated the interplay between critical slowing down, long-term rhythms, interictal spikes, and seizures in humans.

In the present study we report findings supporting the notion that the seizure onset constitutes a critical transition from normal to a seizure brain state. On a small time-scale, two markers of critical slowing down—autocorrelation and variance—showed changes close to the seizure onset which indicated the presence of a critical transition in 13 of the 14 patients (Supplementary Fig. 2). Over a longer time-scale, increases in both signals were observed in most patients (9 of 14 patients) over periods ranging from tens-of-minutes to days (Supplementary Fig. 4). The markers used to detect critical slowing down could be combined with rates of epileptiform spikes to create a powerful forecasting tool. The long-term and continuous nature of the data used in this study was extremely important, as the relationship between seizures and the signals changed over much longer time scales than would normally be obtained in a clinical setting. Our work contributes four major findings to the field:

1. Evidence that seizures onsets involve critical transitions, as suggested by computational models of epilepsy[18,21,36]. This

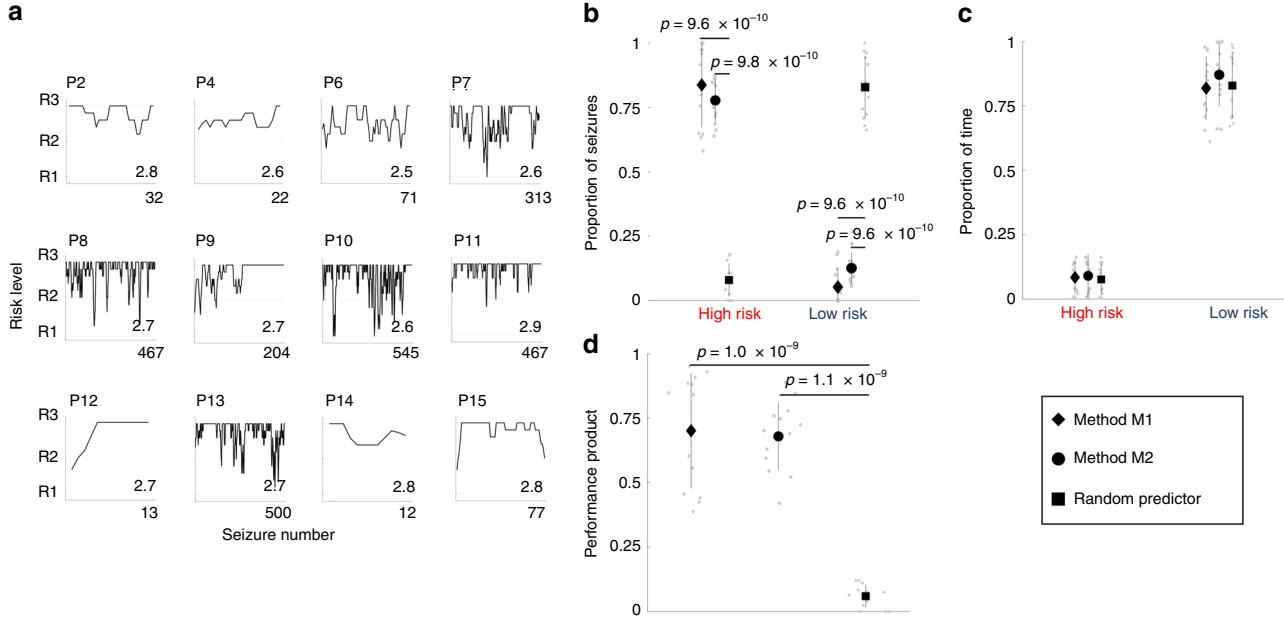

**Fig. 5 Forecasting performance for 14 patients. a** The pseudoprospective approach was applied to all patients, except Patient 5 due to too few seizures. A moving average (length 5 seizures) is shown for each patient. The number inset in each subplot represents the average risk level assigned to seizures. **b** The proportions of seizures correctly classified as high risk or incorrectly classified as low risk are compared for Method M1 (diamond), Method M2 (circle), and a random predictor (square). There was a significant effect of method used to forecast seizures in the high-risk category ($F_{2,12} = 112.3$, $p = 7 \times 10^{-13}$). A post-hoc analysis showed that both Methods M1 and M2 correctly classified significantly more seizures than the chance model ($p = 9.6 \times 10^{-10}$ and $p = 9.8 \times 10^{-10}$ for Methods M1 and M2, respectively). There were no significant differences between Methods M1 and M2 ($p = 0.06$). Methods M1 and M2 had significantly fewer seizures in the low-risk category than the random predictor ($F_{2,12} = 148.3$, $p = 9.6 \times 10^{-10}$ and $p = 9.6 \times 10^{-10}$ for Methods M1 and M2, respectively). There were no significant differences between Methods M1 and M2 ($p = 0.5$). **c** The proportions of time spent in the high and low risk categories for the three methods. No significant differences were found between the models ($F_{2,12} = 1.13$, $p = 0.3$ and $F_{2,12} = 2.12$, $p = 0.1$ for high and low risk categories, respectively). **d** Based on the performance product, there was a significant effect of method ($F_{2,12} = 73.7$, $p = 6 \times 10^{-11}$) and both Methods M1 and M2 performed significantly better than the chance predictor ($p = 1.0 \times 10^{-9}$ and $p = 1.1 \times 10^{-9}$ for Methods M1 and M2, respectively). There was no significant difference between Methods M1 and M2 ($p = 0.2$). Black symbols in plots **b–d** represent population means and lines represent ±one standard deviation. Gray symbols represent raw values for each patient ($N = 13$ patients). Statistical comparisons were computed using a balanced two-way ANOVA corrected with a Tukey–Kramer multiple comparisons test.

provides important validation of the mathematical models used in epilepsy that have so far proved difficult to verify in humans.

2. Changes in the autocorrelation over long time scales were not confined to a localized region in the brain, suggesting that changes in susceptibility are detectable across broad areas of the brain.

3. Seizures tended to occur on a narrow phase of the periodic autocorrelation, variance, and spike rate signals. These signals provide a powerful forecasting tool that can be used to determine seizure susceptibility.

4. Seizures tended to cluster after lead seizures. The period during clusters was also characterized by a sustained increase in the autocorrelation, which is consistent with an increase in overall brain excitability. This result provides a mechanistic basis for seizure clusters.

Critical slowing down has been observed prior to seizures in experimental studies[18,35,37] and at the end of most seizures in humans[15]. Two human studies showed evidence of linear and nonlinear changes in intracranial signals prior to seizures, providing some evidence for seizures as a critical transition[38,39]. In contrast, two recent studies found little evidence of critical slowing down prior to seizures[20,40]. Our results do not contradict these recent observations, but instead suggest that the warning signals fluctuate over longer temporal scales (hours and days) than those regarded in their study (seconds to minutes). Furthermore, our previous work has demonstrated the lack of signal

stability in the weeks following surgical implantation which could contribute to the differences observed[41,42].

Figure 1a illustrates how a bifurcation can lead to critical slowing in the EEG signal[43]. While other bifurcations (mono-stable or multi-stable models) are also plausible, noise-induced fluctuations are expected to increase in intensity near any critical (i.e., second or higher order) phase transition and display the characteristic features of critical slowing down[18]. The example in Fig. 1a may be oversimplified, but it captures an essential aspect of the dynamical changes that are supported by our results. The example demonstrates how the seizure onset can be characterized in terms of the system time constant and how the time constant changes near a critical point. The transition into the seizure state can occur from a steady approach to the critical point, from a strong perturbation that triggers seizures in a probabilistic manner, or from a mixture of both[3,35]. In reality, it is unlikely that the complex repertoire of brain dynamics can be described by a simple one-dimensional model. Figure 1c, d showed evidence of critical transitions occurring within the seizure itself, highlighting the complex nature of seizures and the presence of higher dimensional transitions.

For most patients, both the autocorrelation and variance signals remained high for a prolonged time after a lead seizure, far longer than the duration of the seizure itself (Fig. 3c–e). This led to a state where seizures were more likely to keep occurring, leading to seizure clusters. While the mechanisms for seizure clusters are poorly understood, it has been suggested that an ictal

focus becomes more excitable, or less inhibited following a first seizure[30,31]. This observation is supported by our results, where a high autocorrelation and variance following a lead seizure would suggest that the brain remains in a more excitable state—a state that is less capable to recover from external perturbations.

There is accumulating evidence that suggests seizures are mediated by long cycles[44]. Seizures and subclinical epileptic activity, such as spike discharges, have been found to be distributed into circadian cycles[29,30]. More recently Baud et al. [28] and Karoly et al. [45] found that epileptiform activity and seizures fluctuate with daily and multidien rhythms. The current study builds on these previous analyses by showing that the autocorrelation and variance signals fluctuate with similar rhythms to those found in epileptiform activity, and that these rhythms are closely linked to seizure likelihood. Furthermore, the rhythms related to critical slowing down were detectable across most electrodes. This was in contrast to the rates of epileptiform activity, which tended to be localized to specific electrodes (Fig. 3b).

In this study, two methods for seizure forecasting were explored: Method M1, where seizure rhythms were calculated using all available data, and Method M2, where seizure probabilities were iteratively computed based on past seizure occurrences. Both methods could accurately forecast seizures (average sensitivity $84 \pm 16\%$ and $77 \pm 8\%$, respectively), performing significantly better than chance (9.6%). In the original study[24], $72 \pm 13\%$ of seizures were correctly classified as high risk during the training phase. However, the performance dropped to $58 \pm 25\%$ during the advisory phase (Supplementary Table 1). The percentage of time spent in high risk in the original study was greater than in the current study: $31 \pm 8\%$ and $25 \pm 10\%$ during the training and advisory phases, respectively, cf. $8 \pm 6\%$ and $9 \pm 8\%$ using Methods M1 and M2, respectively. It should be noted that the original trial used only a subsample of the data used in the current study.

Here, we have applied a theoretical approach, critical slowing down, to the forecasting of seizures. This approach outperformed all previous attempts to predict seizures on the same dataset (Supplementary Table 2). It is important to note that previous studies have not made use of the longer rhythms. The longer rhythms are most likely the main contributor to the improved outcomes as the long cycles could be used to greatly reduce the time in high. Furthermore, it is important to note that the time in high risk was defined differently for the various studies, hence they may not be directly comparable. Our approach has been designed to give a warning at the sample prior to a seizure (i.e. 2–4 min prior to a seizure).

Seizures occurred in a patient-specific and probabilistic manner relative to the two warning signals. Forecast performance significantly improved when the combination of autocorrelation, variance, and spike information was included compared to using spike rates alone (Supplementary Fig. 21), demonstrating the power of using additional predictors. The need for combining statistical priors is a framework that is being accepted in the seizure forecasting community and new methods for combining multiple predictors are emerging[2,34]. The future of seizure forecasting will undoubtedly include multi-modal information, combining a patient-specific mixture of implanted and wearable technologies[1,2,46].

The underlying mechanisms that modulate seizure susceptibility remain unknown. A few patients had ~12 or ~24-h cycles, which could be influenced by hormonal fluctuations, such as changes in cortisol and melatonin[47,48]. Anti-epileptic drugs play an important role in modulating cortical excitability[49] and, thus, likely influence the patterns observed in this study. However, Patient 6 was not on medication, yet had a strong circadian cycle,

demonstrating that the circadian influence on seizures was not modulated by anti-epileptic drugs in this patient. The multidien cycles observed for most patients highlight the presence of slow variables (>24 h) that influence seizure susceptibility[28,45,50]. The causal factors of these slower cycles may be regulated by the body[51] or relate to external factors such as weather[52] or behavior[53]. Identification of the causal factors will undoubtedly improve our understanding of seizures and improve techniques that make seizures predictable.

Seizure forecasting has the potential to transform the clinical approach to the treatment of epilepsy. An accurate forecast could be used to provide the patient a warning and also trigger interventions. It is clear from this, and other recent studies[28,45] that long-term monitoring in epilepsy will be necessary to create patient-specific clinical treatments. A clinical device could be used to intervene during periods of high risk by, for example, applying deep brain stimulation as required. Future interventions that incorporate forecasting will undoubtedly pave the way towards improved outcomes for people living with epilepsy.

## Methods

**Human data.** A total of 15 patients with focal epilepsy participated in the first-in-human study (for patient details, see Cook et al.[24]). All data were collected with ethics approval from Human Research Ethics Committees at the participating institutes. The seizure advisory system captured continuous iEEG recordings on 16 electrodes at 400 Hz sampling rate (Fig. 2a). Patients were implanted with 16 electrodes placed near the presumed epileptogenic zone. A board-approved epileptologist reviewed each patient's iEEG and annotated the seizures and their durations. Only clinically correlated and clinically equivalent seizures, as defined by Cook et al.[24], were considered.

Epileptiform spikes in the data were detected using a correlation-based algorithm that compared the iEEG signal to a patient-specific template which has been described and benchmarked previously[29]. In short, ~100 candidate epileptiform spikes were detected and verified by a board approved epileptologist, individually for every electrode in every patient. The average waveform was used as a template to automatically detect epileptiform spikes. Spikes were detected by computing the correlation between a sample iEEG and the template. Sections of iEEG with a correlation above 0.85 were considered new spikes. Example spikes for Patient 1 are shown in Fig. 6a.

**Data selection and pre-processing.** All analysis and data manipulation were conducted using MATLAB (MathWorks 2017a). Three second snapshots of iEEG recordings separated by 2 min were obtained on each channel. The 3 s segments were filtered using a low-pass filter (finite impulse response filter) with a cutoff of 170 Hz. This was implemented to remove an ~200 Hz artifact in the data that appeared when the patients charged their devices. After filtering, the first and last second of data was discarded, leaving 1 s snapshots of iEEG data separated by 2 min (Fig. 2a). In each segment, the signal variance (Eq. (2)) and autocorrelation (Eq. (3)) were computed

$$V_y = \frac{1}{T} \sum_{t=1}^{T} (y_t - \bar{y})(y_t - \bar{y}) \qquad (2)$$

$$C_\lambda = \frac{1}{T} \frac{\sum_{t=1}^{T-\lambda} (y_t - \bar{y})(y_{t+\lambda} - \bar{y})}{V_y} \qquad (3)$$

Here, $T$ represents the number of samples in the signal $y$, and $\bar{y}$ represents the signal mean. $C_\lambda$ represents the autocorrelation function of signal $y$ as a function of the lag value $\lambda$. The autocorrelation measure used in the study was taken as the width at the half maximum of the autocorrelation function (Fig. 2b). We also investigated the lag 1 autocorrelation measure[20] and found that it gave approximately similar results. However, the autocorrelation width produced a larger dynamic range of values from which to observe changes in the signal. Two examples of the autocorrelation function are shown for two seconds of iEEG in Fig. 6b.

After computing the autocorrelation, variance, and spike rates for the entire dataset, a causal moving average filter was applied with a window of 2 days (1440 samples) to identify long rhythms in the data. The filtered data was then subtracted from the unfiltered data to identify short rhythms. This signal was then smoothed using a causal moving average filter of length 20 samples (40 min). A Hilbert transform (MATLAB's *hilbert* function) was applied to the long and short rhythms to compute the analytic signal, from which signal phases could be derived. All results, unless otherwise stated, used causal moving average filters.

The autocorrelation and variance signals were also computed in a 5 s moving window with half a second overlap between windows. These were computed from

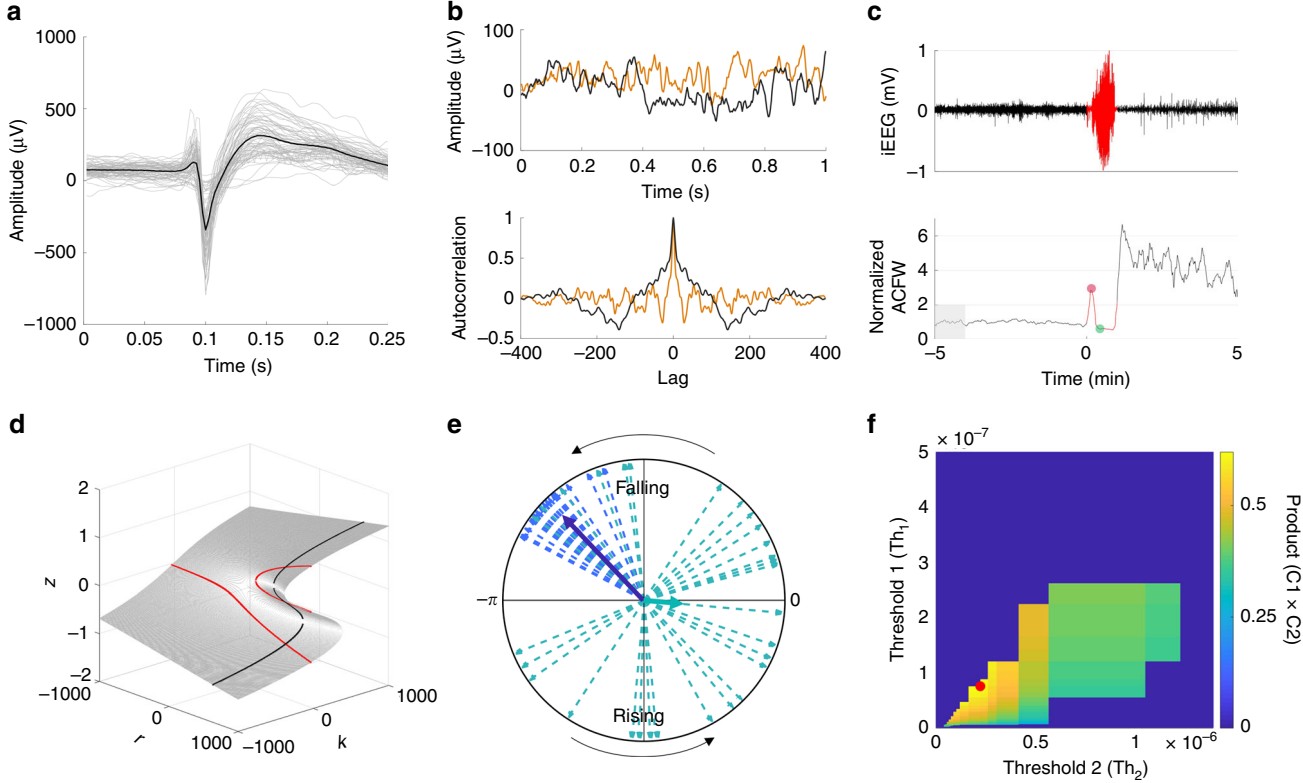

**Fig. 6 Methods used in the analyses. a** Example of epileptiform discharges (termed 'spikes' throughout the text) detected using a patient-specific template matching algorithm. **b** Two examples of 1 s length iEEG signals (top) and the corresponding autocorrelation function. **c** An example iEEG signal during a seizure (top) and the corresponding ACFW (bottom). The red region shows the seizure. The ACFW was normalized by the average of a 1 min period prior to the seizure onset (shaded region). **d** The example system used throughout the text has equilibria given by a manifold (gray surface). The parameters $k$ and $r$ represent two driving parameters that control the state of the system, and how susceptible it is to a state transition. **e** The synchronization index (SI) is computed by adding complex phases (dashed vectors) and computing the mean resultant length (bold vectors). Shown are two examples, one with a high SI (dark blue) and one with a low SI (green). **f** Two thresholds were optimized to separate seizure probability into low, medium, and high risk states by maximizing the product of criteria C1 and C2. Regions where criteria C3 and C4 were not achieved were set to zero. In this example, the red dot shows the optimal thresholds where C1 and C2 were maximized.

3 h prior to and after each seizure (i.e. Fig. 1c, d and Supplementary Fig. 2). These data were used to investigate critical slowing down and evidence of state transitions on a fine temporal scale. The changes in ACFW for all seizures were compared across patients. Due to the variability across seizures and patients, the ACFW was first normalized by a baseline period 5 min prior to the seizure onset (Fig. 6c). The ACFW peak (red dot) and subsequent trough (green dot) were then computed and compared for all seizures across all patients (Supplementary Fig. 2).

**Descriptive model.** Throughout the text we give examples of a bi-stable dynamical system and use it to predict the changes in system time-constant and auto-correlation that should be seen at a critical transition (Fig. 1a and Supplementary Fig. 5). The model we use is given by the following differential equation:

$$\frac{dz}{dt} = -z^3 + (1 \times 10^{-3})rz + (1 \times 10^{-3})k. \qquad (4)$$

The equilibria of this equation describes a manifold as shown in Fig. 6d. By fixing the value of $r$, the bi-stable system from Fig. 1a can be generated (black line). For a fixed value of $k$, the system shown in Supplementary Fig. 5A can be generated (red line). To generate the system in Fig. 1a we set $r = 500$ and varied $k$ over ±200. To compute the system time constant, we linearized Eq. (4) by evaluating the Taylor series and solving for $z$. This model is a purely descriptive system and other dynamical systems, such as mono-stable or multi-stable systems, are also plausible. However, this model provides an intuitive way to analyze what occurs near a critical point, then test any predictions in the data. The effects close to the critical point generalize for a wider class of models that describe a critical transition.

**Missing data.** Over the course of the study, many data dropouts occurred when the recording system was not fully recharged or when data was not regularly retrieved. As a result, all patients had gaps in their data, lasting from minutes to days. In most cases, the dropouts were short segments. In one case (Patient 3), the segments of dropout data accounted for almost 80% of the total recording duration, so this patient was removed from analysis. With Patient 3 removed, gaps in the data comprised 26% of the total data (minimum: 1.25%, maximum: 40.11%; Table 1).

Short sections (<2 h) in the autocorrelation, variance, and spike rate signals that contained dropouts were filled with Gaussian noise. The mean and standard deviation of the noise was computed from the remaining data without dropouts. Sections with larger gaps were left as missing values. These missing values were ignored when computing averages.

When computing the Hilbert and the FT, missing data were first filled with Gaussian noise (as with the shorter sections). This has the effect of introducing noise into the FT and the Hilbert transform. When computing the signal phases from the analytic signal, 60 samples (2 h) either side of dropouts were removed from analyses to reduce the effect of boundaries.

**Synchronization index (SI).** The phases at the times of the seizures (Fig. 2c, red triangles) were used to calculate the SI[54]. Each phase, given by the analytic signal derived from the Hilbert transform, is represented by a complex number that can be drawn on polar axes as a vector (Fig. 6e, thin arrows). The SI is given by

$$\text{SI} = \left| \frac{1}{N} \sum_{n=1}^{N} X_n \right|, X_n = e^{i\theta_n} \qquad (5)$$

where $X_n$ represents the complex-valued analytic signal (magnitude omitted) of the autocorrelation, variance, or spike rate signals at the sample number $n$, and $\theta_n$ is the phase of the signal. $i = \sqrt{-1}$ and $N$ represents either the total number of seizures, or the length of the signal depending whether the SI was being used to compute the seizure histograms, or the phase uniformity of the signal (Fig. 2d, e). If all seizures occur at nearly the same phase of the filtered signal, then the SI will be close to 1 (e.g. dark blue vector in Fig. 6e). If seizures occurred on random phases of the filtered signal, the SI will be close to 0 (e.g., the green vector in Fig. 6e). The SIs reported in this study were computed at the sample prior to each seizure.

**Similarity between electrodes**. The similarities between the three signals across electrodes were compared using a correlation coefficient. The autocorrelation, variance, and spike rate signals were first smoothed using a causal moving average filter of length 20 and sections containing missing values were removed. A cross-correlation (MATLAB *corrcoef* function) was then computed between the signals on each electrode. The cross-correlation produces a 16 × 16 matrix representing the correlation between each electrode combination. The matrix was transformed into a vector with the diagonal and duplicate values excluded, and the mean correlation was then computed and compared (Fig. 3a).

**Seizure clusters**. Seizure clusters were determined by analyzing the inter-seizure intervals for each patient. We plotted a histogram of seizure intervals with a bin spacing of 1 h. Patients that did not have at least five seizures within an interval of 1 day (Patients 2, 4, 5, and 14) were not considered to have seizure clusters. For the remaining patients, we used the histogram to determine the seizure lead times. The histograms showed two types of responses: (1) there was an exponential decay from time zero or (2), there were multiple peaks at regular intervals. For example, Patient 1 had a peak close to zero and an exponential decay without any other obvious peaks. Patient 9 had multiple peaks at daily intervals (Supplementary Figs. 6 and 13). For cases where there was an exponential decay, we set the lead time to be 1 day. For patients where there were multiple peaks, we set the lead time to the first trough between peaks. For example a lead time of 0.6 days, or 14.4 h was chosen for Patient 9. Seizure clusters were investigated relative to the autocorrelation and variance signals on the channel with the highest SI.

**Seizure forecasting**. Seizure risk was computed for each patient using a probability distribution of seizures relative to signal phase. The resulting probability density was used to compute seizure probability over time, from which three risk levels were determined: low risk, medium risk, and high risk. The risk levels over time were computed by thresholding the seizure probability such that the following criteria were optimized:

C1. Maximize the time spent in low-risk periods.
C2. Maximize the number of seizures classified in high-risk periods.
C3. The time spent in low risk is greater than time spent in medium risk. The time spent in medium risk is greater than time spent in high risk
C4. The number of seizures occurring during low risk is less than the number occurring during medium risk. The number of seizures occurring during medium risk is lower than the number occurring during high risk.

The optimization was conducted by maximizing the product of C1 and C2. For Method M1, the product of C1 and C2 at points where C3 and C4 were not satisfied was set to zero. For Method M2, setting these points to zero often resulted in no optimal solution being found, hence optimization was only conducted on C1 and C2. Since the search space was small, the thresholds could be optimized quickly using a brute-force approach. Figure 6f shows the search space for Patient 1. Threshold 1 (Th$_1$) corresponds to the threshold separating the low and medium risk states. Threshold 2 (Th$_2$) separates the medium and high risk states. The combinations of thresholds that best achieved the above criteria are shown by a red circle.

**Method M1**. The potential to forecast seizures was evaluated using the short and long rhythms of all three signals (autocorrelation, variance, and spike rate) over all the data. The probability of a seizure given phase was computed using the phase estimated from the analytic signal. The phases between $-\pi \leq \theta < \pi$ were broken up into 20 equally spaced windows. A probability given phase was computed by evaluating the number of seizures that occurred in a phase window ($S_\theta$) divided by the number of times the phase appeared in the signal ($N_\theta$):

$$P(S|\theta) = \frac{S_\theta}{N_\theta} \qquad (6)$$

The probability density for the combined short and long rhythms, and for the combined autocorrelation, variance, and spike rate signals were computed by multiplying the probabilities together.

**Pseudo-prospective seizure forecast (Method M2)**. Forecasts using Method M2 employed causal filtering to identify the short and long rhythms, and to estimate the phase relationships between seizures and the signals. The risk level for each seizure was determined iteratively using the seizure rhythms and risk level thresholds that were determined from past seizure information bootstrapped using data from the first 10 seizures. When a new seizure occurred, the relationship between seizures and the signal phases were recalculated and used to re-estimate the seizure probability distribution, which remained fixed until the next seizure. Due to non-stationary effects in the signals[42], the seizure probability distribution was calculated over a 50-day window.

**Random predictor**. The performance of the two forecasting methods were compared to a random predictor using a random Markov model. To compute the random model, the transition probabilities for each risk state were first computed (based on Method M1); i.e., the probability of transition from low risk to medium risk, medium to high risk, etc. Then, a model that randomly transitioned between the three risk states was then generated using the transition probabilities. Statistical

differences in the data were computed using ANOVA followed by post-hoc analysis using Tukey–Kramer comparison, where necessary, with $\alpha = 0.01$.

**Reporting summary**. Further information on research design is available in the Nature Research Reporting Summary linked to this article.

## Data availability
Seizures and some segments of the data used in this study are currently publicly available on the online platform Epilepsyecosystem[34]. Other segments of the data can be made available upon reasonable request.

## Code availability
Code, sample data, and examples to generate some figures used throughout this paper can be found at https://github.com/matiasim/Critical_Slowing_Epilepsy.

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

## Acknowledgements
IEEGPORTAL and National Health and Medical Research Council Project Grant 1130468. C.M. acknowledges support from a NARSAD Young Investigator grant. M.I.M. acknowledges support from the Melbourne Neuroscience Institute Fellowship award. D. R.F., L.K., and M.J.C. acknowledge the support from the Epilepsy Foundation of America, My Seizure Gauge Challenge Funding. P.J. and J.H. acknowledges support from the Ministry of Health of the Czech Republic 17-28427A.

## Author contributions
Conceived and designed the work: D.R.F., C.M., L.K., M.J.C., M.I.M., P.J.K., P.J., J.H., J.K. Performed the analysis: M.I.M., C.M., D.R.F., L.K. Wrote the paper: M.I.M., C.M., K.D., W.D., D.B.G., A.N.B., M.J.C., L.K., D.R.F.

## Competing interests
The authors declare no competing interests.
