## [Peer Review File · Nature Communications]

Reviewers' comments:

Reviewer #1 (Remarks to the Author):

Authors reported an interesting study related to critical slowing as a reliable indicator that could be used in seizure forecasting algorithms

The study is novel and significant to the field for many neuroscientists working on this area.

My main comments focus on improving further the details of the preprocessing steps in order to clarify every aspect of the research.

1) Authors should report any basic preprocessing steps of the original iEEG. denoising, filtering (type/order of the filter etc) frequencies

2) Authors analyzed iEEG in two temporal scales: How did you define short and long scales ?

The long rhythms were extracted by applying a moving average filter with a length of 2 days to the signals. Which was the length of moving average filter ?

The short rhythms were extracted by subtracting the long rhythm from the raw signals and smoothing with a moving average filter of length 20 time-units.

It is important to report the procedure of defining short and long rhythms in detail.

3) Two thresholds that optimally separated the low, medium, and high-risk categories were computed and used to categorize risk state over time (Example Figure 4B, bottom)

How did you detect the two thresholds per subject ?

4) Which iEEG from the 16 were the more informative related to the performance of M1 and M2 methods.

Did you detect any variability across sensors and also across patients ?

Reviewer #2 (Remarks to the Author):

Maturana and colleagues have suggested, based on iEEG data, that critical slowing down is observed in epilepsy patients, and can be associated with seizure forecasting. They examined the data from 14 patients, and measure three indices of the time series. They show that in some patients the seizures occur predominantly during specific phases of the cycles of these indices, and they argue that these indicate a signature of phase transitions in the brain.

There are no proper justifications for a discontinuous bifurcation, that has a bistable region, to propose this framework to analyse the data.

There are several confounds on the indices used to detect critical slowing down, or the distance to criticality. The seizures themselves cause an increase in the indices used to seizure forecasting. In this sense, it is not possible to know whether the seizures tend to occur at specific phases or whether seizures cause the indices to be usually at specific phases.

The manuscript should be more self-contained and not rely on so much on previous papers. In these cases, at least a brief explanation should be provided. For example, a description of the correlation algorithm used for patient-specific spike should be given.

The results section contains too much detail that should be presented in the Methods.

On page 7, spikes are defined as a rate; in page 8, it is used as spike rate. This terminology is confusing.

It seems that a proper ROC curve is missing to interpret the results and the likelihood of false positives and false negatives.

Reviewer #3 (Remarks to the Author):

Maturana et al used three aspects of iEEG time series to perform seizure predictions on long-term recordings from 14 patients. They considered autocorrelation times (T), variance (V), and interictal spike rate (R). They found that changes in T, V, and R on two time scales (40 min and 2 days) were related to changes in seizure probability. Typically, they found that seizure probability was highest on the up swing of T, V, and R.

The authors interpret these results as support for the hypothesis that the transition to a seizure state is a phase transition (in the sense of statistical physics) or bifurcation (in the sense of dynamical systems theory). For many phase transitions and bifurcations, it is predicted that T and V should increase and reach a maximum and then decrease as one crosses the tipping point of the transition. This peak in T at criticality is called "critical slowing down". Thus, they interpret the rise in T and V as evidence for critical slowing down and they interpret the concurrent rise in seizure probability as evidence for the hypothesis that a seizure entails a phase transition (or bifurcation).

The less controversial part of the authors work is the part about improving seizure predictions using T and V. This part alone would be a somewhat incremental advance on previous work (the authors cite many papers on seizure prediction). In contrast, the more controversial, innovative, and interesting parts of the work are the claims related to criticality and critical slowing down. However, my primary concerns are related to the claims about criticality and critical slowing down.

MAJOR CONCERN 1: The following concern is acknowledged by the authors in their discussion, but I'm not convinced that they address it in a satisfactory way. The data indicate that peak T and V do not coincide with peak seizure probability. Instead, peak seizure probability was on the rising phase of T and V, preceding peak T and V by a day (estimated by the detailed data for Patient 1 in Fig 2). This observation seems to be inconsistent with the hypothesis that transition to the seizure state is a critical phase transition. The authors explain on pg 22 "This is most likely because the seizure event itself is characterized by an even larger increase in autocorrelation and variance, which confounds the analysis". However, this explanation is inconsistent with the criticality and critical slowing down hypothesis. More specifically, if the seizure state is on one side of the phase transition and the non-seizure state is on the other side of the phase transition, then T should peak at the boundary between these states, not on the seizure side of the boundary. According to the authors hypothesis, T should not be highest in the seizure state. This leads me to conclude that the authors data invalidates their own hypothesis. The data certainly support that T and V are

related to seizure probability, but do not seem to support the hypothesis that the underlying mechanism is a critical phase transition. But, without the interesting hypothesis about criticality to back it up, the relationship between T, V, and seizure probability does not seem sufficiently interesting to warrant publication in Nature Communications. Therefore, if the authors cannot revise their hypothesis in some way to make it consistent with the observations, I am hesitant to endorse publication.

MINOR CONCERNS

1. (related to MAJOR CONCERN 1) For all patients, except Patient 1, it is not possible to relate a phase lag of T and V to an actual amount of time passed. This makes it difficult to assess the results. Results would be easier to assess if the reader could know the actual time delay between peak seizure probability and peak T or V (for all patients).
2. (end of pg 3) The list of cited contexts for seizures and phase transitions should include pharmacologically induced phase transitions. For example: Gautam, S.H., Hoang, T.T., McClanahan, K., Grady, S.K., and Shew, W.L. (2015). Maximizing Sensory Dynamic Range by Tuning the Cortical State to Criticality. PLOS Comput. Biol. 11, e1004576.
3. (middle of pg 4) The concept of "control parameter" is not likely to be understood without a definition for the readers. Moreover, "effective connectivity" is not a control parameter. The real, anatomical connectivity and synaptic efficacies are control parameters. "Effective connectivity" is more like an order parameter, based on measure dynamics.
4. (Fig 1) The cartoon example is a hysteretic bifurcation. However, a more common paradigm for studying critical phase transitions (e.g. in the Ising model or directed percolation) is a continuous phase transition (which has no bistable solutions). Continuous phase transitions also exhibit critical slowing down and are simpler to understand. It is not clear to me why the authors are using the more complex example scenario to illustrate their hypothesis.
5. (Fig 1) Are the data in panel B actually from integrating some particular differential equation? If so, this must be described somewhere (in methods perhaps?). The vertical axis in B should be labeled.
6. (middle of pg 6) The authors discuss eigenvalues and other math that is not likely to be understood by readers without further explanation. At the very least, the authors should cite a reference that would help an interested reader learn about the relevant math.
7. (last paragraph pg 9) It would be helpful to explicitly state that your SI has nothing to do with synchrony of brain activity (e.g. two electrodes with synchronous fluctuations).
8. (Fig 2B) Is this a real autocorrelation function from your data? If not, it would be helpful to see real examples. Perhaps example cases with large T and small T would be most helpful. The shape of these functions would help clarify whether oscillatory activity or more noisy activity is responsible for different T values.
9. (top pg 11, relates to Major Concern 1) Here it is claimed that, for four patients, the sleep/wake cycle dominates the changes in T and V, which is plausible. It is also claimed that the sleep/wake transition is also a critical transition. If these are both true, then the author's hypothesis would suggest that seizures should be most prevalent at night. Why is this not the case. A more convincing argument is needed to make the hypothesis consistent with the data.
10. Some kind of report of a "false alarm" rate should be given for the various seizure prediction methods. I understand that the performance metric is a step in this direction, but a more direct report of false alarm rates is needed.

11. (middle of pg 18) "combed approach" should be "combined approach", I assume.
12. (pg 19, point 1) The authors need to cite some of the model/theory that you claim your work supports. If none exist to cite, then remove the point.
13. (middle pg 20) The authors state "The current study builds on these previous analyses by showing that seizures arise as the brain approaches a critical transition..." This is stated too strongly. The authors actually hypothesize that seizures arise as the brain approaches a critical transition. The authors have not shown this.

Reviewers' comments:

Reviewer #1 (Remarks to the Author):

Authors reported an interesting study related to critical slowing as a reliable indicator that could be used in seizure forecasting algorithms. The study is novel and significant to the field for many neuroscientists working on this area.

My main comments focus on improving further the details of the preprocessing steps in order to clarify it every aspect of the research.

1) Authors should report any basic preprocessing steps of the original iEEG. denoising, filtering (type/order of the filter etc) frequencies

There was a filter applied to remove a high frequency artefact that appeared when patients charged their device. This was not originally stated. The filter was a finite impulse response low pass filter with cutoff at 170 Hz. This has now been included in the methods section (lines 535 – 541). No other pre-processing steps were conducted outside those already stated in the methods.

2) Authors analyzed iEEG in two temporal scales:
How did you define short and long scales?

The two temporal scales were chosen after carefully studying the data and noticing that all patients had prominent rhythms of 0.5-1 day, and some patients had longer rhythms of 3 days or greater. Furthermore, these longer rhythms had been identified previously in other work by our group (Karoly 2018, Lancet Neurol). These periods were identified by analyzing the FT of the autocorrelation and variance signals (Supplementary Figure 3). Based on this analysis, the short rhythms were defined as anything with a period of less than 2 days, and long rhythms as anything greater than 2 days. This is also in line with observations from the NeuroPace group (ref [35]) who found multidien rhythms were predominantly between 7-26 days. In the revised version of the manuscript, we have added a line in the text to make this clearer (lines 212 – 220).

The long rhythms were extracted by applying a moving average filter with a length of 2 days to the signals. Which was the length of moving average filter?

The short rhythms were extracted by subtracting the long rhythm from the raw signals and smoothing with a moving average filter of length 20 time-units.

The length of the 2-day moving average filter was 1440 samples. The length of the smoothing filter was 40 mins, or 20 samples. We have clarified this in the relevant section (lines 212 – 220).

It is important to report the procedure of defining short and long rhythms in detail.

The existence of distinct short (about daily) and long (multiday) rhythms has been established in prior literature. In this study we sought to apply a generalized method to relate the phase of these rhythms to signatures of critical slowing down. We defined the short and long rhythms after noticing in the Fourier transform that most peaks were at periods 1 day or less, and in some cases 3 days or more. By separating the data into these two temporal scales, we could explore (through the Hilbert transform), the relationship between seizures and any rhythms with periods less than 1 day, or greater than 2. This has been clarified in the appropriate section (lines 212 -220). Separating the data into these separate timescales is important as computing the Hilbert transform on the raw data would not be possible (i.e. it is not possible to the phase of a wideband signal).

We have chosen to analyze pre-defined short and long rhythms with a cut-off of 2 days (i.e. <2 days and >2 days). It is possible that forecasting results could be improved by choosing patient specific cut-offs, however, patient specific cutoffs were beyond the scope of this study.

3) Two thresholds that optimally separated the low, medium, and high-risk categories were computed and used to categorize risk state over time (Example Figure 4B, bottom)

How did you detect the two thresholds per subject?

The thresholds were derived from the seizure probability (as calculated from the phase of autocorrelation, variance and spike rate rhythms). We have clarified this in the main text (lines 328-330):

“Using the seizure probability described above, two thresholds that optimally separated the low, medium, and high-risk categories were computed and used to categorize risk state over time (Example **Error! Reference source not found.**B, bottom). Details on how these thresholds were set are described in the Methods.”

The methods section, “Seizure Forecasting” contains more extensive details on how the thresholds were optimized (section starting Line 638).

4) Which iEEG from the 16 were the more informative related to the performance of M1 and M2 methods. Did you detect any variability across sensors and also across patients?

This is a really interesting question. We didn't specifically test the prediction performance of each individual electrode. However, we were careful about how we chose which electrode to use. The electrode we used for prediction was the electrode that had the highest synchronization index (termed SI in the text). This means that the chosen electrode was the one that was able to best detect the relationship between seizures and the underlying rhythms. The assumption here is that the other electrodes would have performed worse.

In terms of the variability across electrodes, we tested this by looking at the variability of the SI across electrodes. Visual inspection showed that the autocorrelation signal was similar across most channels in most patients. However, the spike rate tended to be quite different across channels. To test the similarity, we computed the correlation between the autocorrelation signal across all pairs of electrodes. This was repeated for the spike and variance signals. Figure 3B shows the mean across all possible pairs for each patient. Importantly, what it shows is that the mean cross correlation for the autocorrelation signal was high, suggesting that there is little variability across channels. In contrast, the mean cross correlation across channels for the spike rate signal was low, suggesting that there was a large amount of variability across channels.

This analysis therefore may suggest that the autocorrelation is indicative of brain-wide (or at least over the recorded regions) changes. In contrast, the spike rate signal is descriptive of local changes. This is now discussed in the discussion section (lines 504-512)

Reviewer #2 (Remarks to the Author):

Maturana and colleagues have suggested, based on iEEG data, that critical slowing down is observed in epilepsy patients, and can be associated with seizure forecasting. They examined the data from 14 patients, and measure three indices of the time series. They show that in some patients the seizures occur predominantly during specific phases of the cycles of these indices, and they argue that these indicate a signature of phase transitions in the brain.

There are no proper justifications for a discontinuous bifurcation, that has a bistable region, to propose this framework to analyse the data.

The motivation for a bistable dynamical model comes from various models of epilepsy and seizures which suggest there is a state transition that occurs during seizures. In the text we give reference to various models that would suggest this (lines 72-74):

“It has been hypothesized that a rapid transition from normal brain activity to an epileptic seizure is such a critical transition [3, 4, 18-21].”

From a modelling point of view, multiple pathways to a seizure are possible. For example, Jirsa 2014 (reference 23 in the text) looks at the various bifurcation topologies possible and concludes that four topologies for seizure onset are plausible, all of which consist of a critical transition. Not all topologies are bistable, some are monostable. However, regardless of the type of transition, nearly all the models will exhibit changes in the autocorrelation signal when a critical transition is crossed. Even models that transition into a limit cycle will exhibit a drop in the autocorrelation, assuming that frequency of the limit cycle is fast enough (this is a fair assumption in epilepsy). The model we present is a way of conceptualizing the critical transition and particularly, what happens at the transition point, rather than what happens during the seizure itself.

To clarify this, we have re-written the Results – conceptualization of critical slowing down in epilepsy. We have now modelled the critical transition in terms of the system time-constant and shown that all seizures show strong evidence of a critical transition. Assuming that linear stability analysis applies as described in the text, a large class of models will show the characteristic features of critical slowing down, and the transition from one state to another will show a sharp drop in autocorrelation. We have also further addressed this in the discussion (lines 420-428).

There are several confounds on the indices used to detect critical slowing down, or the distance to criticality. The seizures themselves cause an increase in the indices used to seizure forecasting. In this sense, it is not possible to know whether the seizures tend to occur at specific phases or whether seizures cause the indices to be usually at specific phases.

This is true, it can be difficult to know if a change in any of the signals was going to occur regardless of the seizure, or if the seizure caused a change. Overall, seizures represent a very small amount of the total data in most cases and it is unlikely that the periodic signals are driven purely by the seizures. Some evidence for this is the strong presence of both short and long cycles even in patients with very few seizures. This is quantified by a very low SI when computed across the signals (rather than seizures) - demonstrating a uniform distribution of phases over time - for almost every patient. You can see Patient 4 (Supplementary Figure 8) as an example. Further evidence for this is the presence of the very strong daily rhythm in Patient 1 (figure 2C) despite no seizures occurring on most days.

The primary confounding factor is likely to be the sleep-wake cycle, which causes a large change in the signals tracked for this study. However, what is encouraging is that seizures can be accurately predicted by analyzing the relationship between the three signals (autocorr, spikes and variance) and seizures. The pseudo-prospective approach

(Method M2), where at each time point a future prediction is made based on past data, also performs well. This suggests that the predictive relationships are strong. However, the data does little to reveal any mechanistic insights as to what is causing a change in any of the signals.

The reviewer's comment regarding distance to criticality is an important one. Perhaps we put too much emphasis on critical slowing down with regards to seizures. It is likely that critical slowing down is also happening at a transition from wake to sleep and throughout various sleep stages. Furthermore, specific brain states, such as sleep, cause large changes in the autocorrelation (or other signals; See Supplementary Figure 24 for an example).

For these reasons, we have divided the manuscript to be focused critical slowing down on a very short time scale, and then separately looked at how these features change on a long time scale. On a short time scale, we have provided new evidence that seizures do in fact constitute a critical transition. On a long time scale we have re-worded the text to give less emphasis on critical slowing down, and more emphasis on the relationship between the rhythms and seizures.

The manuscript should be more self-contained and not rely on so much on previous papers. In these cases, at least a brief explanation should be provided. For example, a description of the correlation algorithm used for patient-specific spike should be given.

A description of the correlation algorithm has been added. We have expanded on the methods sections to also make each section clearer.

The results section contains too much detail that should be presented in the Methods.

We have reworded the results and put more into the methods section.

On page 7, spikes are defined as a rate; in page 8, it is used as spike rate. This terminology is confusing.

We have reworded all relevant sections to say "spike rate"

It seems that a proper ROC curve is missing to interpret the results and the likelihood of false positives and false negatives.

We have now included ROC curves for both forecasting methods M1 and M2.

We previously did not include it because it can be difficult to interpret and compare to other similar studies. The reason is that with both methods, patients spend a significant

amount of time in low risk. This is mainly from the addition of the long rhythms. Apart from the trivial case where time in high is 100%, the next highest "Time in High" can be very low (see example below). Hence, there tends to be a sharp jump from a low value of TiH to the maximum value. This can also produce a large area under the curve (AUC), a common metric used to compare performance. While this is a good outcome for our forecaster, it is probably not a suitable way to compare different forecasting methods.

Reviewer #3 (Remarks to the Author):

Maturana et al used three aspects of iEEG time series to perform seizure predictions on long-term recordings from 14 patients. They considered autocorrelation times (T), variance (V), and interictal spike rate (R). They found that changes in T, V, and R on two time scales (40 min and 2 days) were related to changes in seizure probability. Typically, they found that seizure probability was highest on the upswing of T, V, and R.

The authors interpret these results as support for the hypothesis that the transition to a seizure state is a phase transition (in the sense of statistical physics) or bifurcation (in the sense of dynamical systems theory). For many phase transitions and bifurcations, it is predicted that T and V should increase and reach a maximum and then decrease as one crosses the tipping point of the transition. This peak in T at criticality is called "critical slowing down". Thus, they interpret the rise in T and V as evidence for critical slowing down and they interpret the concurrent rise in seizure probability as evidence for the hypothesis that a seizure entails a phase transition (or bifurcation).

The less controversial part of the authors work is the part about improving seizure predictions using T and V. This part alone would be a somewhat incremental advance on previous work (the authors cite many papers on seizure prediction). In contrast, the more controversial, innovative, and interesting parts of the work are the claims related to

criticality and critical slowing down. However, my primary concerns are related to the claims about criticality and critical slowing down.

MAJOR CONCERN 1: The following concern is acknowledged by the authors in their discussion, but I'm not convinced that they address it in a satisfactory way. The data indicate that peak T and V do not coincide with peak seizure probability. Instead, peak seizure probability was on the rising phase of T and V, preceding peak T and V by a day (estimated by the detailed data for Patient 1 in Fig 2). This observation seems to be inconsistent with the hypothesis that transition to the seizure state is a critical phase transition. The authors explain on pg 22 "This is most likely because the seizure event itself is characterized by an even larger increase in autocorrelation and variance, which confounds the analysis". However, this explanation is inconsistent with the criticality and critical slowing down hypothesis. More specifically, if the seizure state is on one side of the phase transition and the non-seizure state is on the other side of the phase transition, then T should peak at the boundary between these states, not on the seizure side of the boundary. According to the authors hypothesis, T should not be highest in the seizure state. This leads me to conclude that the authors data invalidates their own hypothesis. The data certainly support that T and V are related to seizure probability, but do not seem to support the hypothesis that the underlying mechanism is a critical phase transition. But, without the interesting hypothesis about criticality to back it up, the relationship between T, V, and seizure probability does not seem sufficiently interesting to warrant publication in Nature Communications. Therefore, if the authors cannot revise their hypothesis in some way to make it consistent with the observations, I am hesitant to endorse publication.

The reviewer raises some well-founded concerns and has provided a really good intuitive way to address the concerns. The reviewer states that: "For many phase transitions and bifurcations, it is predicted that T and V should increase and reach a maximum and then decrease as one crosses the tipping point of the transition." This is correct, the autocorrelation should increase to a peak, then have a sharp drop. This is expected to occur on a short time scale, but not necessarily on the longer time scale as we have used in our paper. The reason for this is because the transition out of the seizure could land you in a different post-ictal state to the pre-ictal state (see the new figure 6D for an example of a possible manifold where this could happen). This claim is supported by our data (Figure 3C), where the autocorrelation signal remains high for a long period (days), likely suggesting a post-ictal state (different from the pre-ictal state) with increased excitability.

We realize that we had put a lot of emphasis on critical slowing and not provided sufficient evidence for it. Hence, we have re-written the first section of the results. We have provided strong evidence that every seizure constitutes a critical transition and shown how, on a short time scale, critical slowing down is observed (Supplementary

Figure 2). For the following sections, we abstracted to longer time scales to show how the metrics used to detect a critical transition can also be used to assess seizure susceptibility. Furthermore, we have included a figure showing the timescales over which we observe an increase in the metrics used to detect critical slowing down (Supplementary Figure 4).

MINOR CONCERNS

1. (related to MAJOR CONCERN 1) For all patients, except Patient 1, it is not possible to relate a phase lag of T and V to an actual amount of time passed. This makes it difficult to assess the results. Results would be easier to assess if the reader could know the actual time delay between peak seizure probability and peak T or V (for all patients).

We have computed the mean cycle time and included this into every phase plot in the supplementary information. We have also summarized the results for the short and long cycles and included it into Supplementary figure 20. In short, the average short cycle across patients was ~ 0.6 days. The average long cycle was ~ 10 days.

2. (end of pg 3) The list of cited contexts for seizures and phase transitions should include pharmacologically induced phase transitions. For example: Gautam, S.H., Hoang, T.T., McClanahan, K., Grady, S.K., and Shew, W.L. (2015). Maximizing Sensory Dynamic Range by Tuning the Cortical State to Criticality. PLOS Comput. Biol. 11, e1004576.

We agree that this is a very interesting and relevant paper. We have now included this in the references

3. (middle of pg 4) The concept of "control parameter" is not likely to be understood without a definition for the readers. Moreover, "effective connectivity" is not a control parameter. The real, anatomical connectivity and synaptic efficacies are control parameters. "Effective connectivity" is more like an order parameter, based on measure dynamics.

What we refer to are other experimental studies from reduced slice preparations which have shown that changes in connectivity or Excitation/Inhibition (E/I) ratio can act as a control parameter governing a phase transition in network dynamics (e.g. Beggs & Plenz 2003, Shew 2009). However, we agree that our terminology is inaccurate. We have removed the phrases with 'effective connectivity'. The section talking about control parameter has been changed to:

While some methods have been developed to track control parameters (variables that drive changes in state) from clinically captured EEG in epilepsy [20, 21], this approach is not straightforward.

4. (Fig 1) The cartoon example is a hysteretic bifurcation. However, a more common paradigm for studying critical phase transitions (e.g. in the Ising model or directed percolation) is a continuous phase transition (which has no bistable solutions). Continuous phase transitions also exhibit critical slowing down and are simpler to understand. It is not clear to me why the authors are using the more complex example scenario to illustrate their hypothesis.

The bistable model was chosen as we believed this would be the most intuitive to understand as the two states are clearly visible. We are introducing this model only to explain the concept but accept that a wider range of models exist that also demonstrate similar characteristics at the critical point. We have now included a section in the methods and supplementary information more clearly describing the example model (lines starting 569). This section should now provide a stronger motivation for why we are using this model as our example.

5. (Fig 1) Are the data in panel B actually from integrating some particular differential equation? If so, this must be described somewhere (in methods perhaps?). The vertical axis in B should be labeled.

The data in Figure 1B (now supp Figure 1B) are simulated responses from an equation that is now given in the methods (Equation 2). This figure should be considered as purely descriptive. We have now also included examples from the data showing how the autocorrelation changes for different signals (Figure 6B).

6. (middle of pg 6) The authors discuss eigenvalues and other math that is not likely to be understood by readers without further explanation. At the very least, the authors should cite a reference that would help an interested reader learn about the relevant math.

We have removed these sections. Instead, we have re-written section 1 of the results to make it more intuitive. We have presented our results in terms of the time constant associated with a linearized system. We have used the Hartman-Grobman theorem (stated in the text) to show what is expected to happen to the system as it crosses a critical transition.

7. (last paragraph pg 9) It would be helpful to explicitly state that your SI has nothing to do with synchrony of brain activity (e.g. two electrodes with synchronous fluctuations). This has been clarified in the text (lines 235 – 237)

8. (Fig 2B) Is this a real autocorrelation function from your data? If not, it would be helpful to see real examples. Perhaps example cases with large T and small T would be most helpful. The shape of these functions would help clarify whether oscillatory activity or more noisy activity is responsible for different T values.

The example in Figure 2B is not real data, it is a descriptive plot. We have now put a real data examples in the Methods section, Figure 6B. This shows two examples, one with a wide autocorrelation function, and one with a thin one.

9. (top pg 11, relates to Major Concern 1) Here it is claimed that, for four patients, the sleep/wake cycle dominates the changes in T and V , which is plausible. It is also claimed that the sleep/wake transition is also a critical transition. If these are both true, then the author's hypothesis would suggest that seizures should be most prevalent at night. Why is this not the case. A more convincing argument is needed to make the hypothesis consistent with the data.

Existing literature would suggest that changes in sleep stages (or other state changes) constitute a critical transition. However, the fact that the autocorr/variance increase at night does not necessarily suggest that seizures should be more prevalent at night. There are two reasons why the autocorrelation might increase irrespective of seizures.

(1) The increase in autocorrelation could be suggestive of a change in the resting state in a dynamical landscape. For example, in Equation (2) and the new Supplementary Figure 4A, decreasing values of r cause an increase in autocorrelation, and move the system away from the bistable regime (unsafe) to a monostable region (safe).

(2) the autocorrelation could increase due to an approach to a critical transition. The critical transition may be a transition into a sleep state, rather than a transition into a seizure state. It is important to note that the example we give in Figure 1 is a simplified 1-dimensional model for descriptive purposes. However, we expect the brain to be of much higher dimensionality, with a multitude of critical points. This higher dimensionality with multiple critical points is highlighted in the new Figure 1C,D, where multiple transitions appear to occur during a seizure. Increases in autocorrelation could therefore represent an approach to a non-seizure critical point, such as a transition occurring during sleep.

We have re-written the first section of the results and relevant sections of the discussion (429 – 440) to make these points clearer.

10. Some kind of report of a "false alarm" rate should be given for the various seizure prediction methods. I understand that the performance metric is a step in this direction,

but a more direct report of false alarm rates is needed.

In response to a similar comment from Reviewer 2, we have included ROC curves for the forecasting. This shows the range of false alarm (or in our case, time in high) vs sensitivity.

11. (middle of pg 18) "combed approach" should be "combined approach", I assume.

This has been corrected

12. (pg 19, point 1) The authors need to cite some of the model/theory that you claim your work supports. If none exist to cite, then remove the point.

The models have been cited

13. (middle pg 20) The authors state "The current study builds on these previous analyses by showing that seizures arise as the brain approaches a critical transition..." This is stated too strongly. The authors actually hypothesize that seizures arise as the brain approaches a critical transition. The authors have not shown this.

We have now re-worked the first section of the results to address this.

Reviewers' comments:

Reviewer #1 (Remarks to the Author):

Authors responded to my comments and with the insertion of new definitions in the main text the readability of the draft has been increased.

I recommend the acceptance of the manuscript without further comments.

Reviewer #2 (Remarks to the Author):

This paper proposes to apply a canonical dynamic systems framework, of low-dimensional systems in the absence of noise, to intracranial EEG in patients with focal epilepsy. It explores the dynamics of a system close to a bifurcation that has a bistable region. A major limitation still remains in the manuscript. The study overlooks important aspects of the dynamics. Presumably, region s_3 (of Figure 1A) should exhibit oscillatory dynamics, which, in the absence of noise, would lead to a periodic autocorrelation. Then, Figure 1B becomes problematic because ACFW is not meaningful in this oscillatory regime (for s_3), and the diagram is misleading. Conceptually, noise is the key element to regulate transitions between stable branches. Its role must be very carefully considered. At the end, the differences between ACFW measured at s_1 and s_2 are very subtle, and the detection of transition to the other branch is trivial, and does not require any index. It seems that critical slowing down is relevant for continuous transitions, and not necessarily for bistable systems in the presence or absence of noise.

Reviewer #3 (Remarks to the Author):

The authors have substantially improved their paper. They effectively resolved all of my minor concerns. Regarding my Major Concern 1 (from first round of review), the authors have made good progress but have not yet sufficiently resolved this concern. To be clear, I will restate and slightly revise Major Concern 1:

Major Concern 1 (revised) A key part of the authors' work is their claim that they have new experimental support for the hypothesis that each seizure is a critical phase transition. In my view, this is the most interesting part of their findings. The additional work about long time dynamics and predicting seizures is also interesting, but it is not sufficiently ground-breaking to merit Nat Comms publication without the new support for the hypothesis that each seizure is a phase transition. Therefore, in my view, it is essential that the authors establish strong, statistically rigorous support for this hypothesis. In their revised manuscript the authors have made good progress towards this, but have not yet sufficiently done so.

They have clarified some previously-confusing concepts. It is now clear that they consider the phase transition to be a fast phenomenon that occurs for each seizure onset (rather than a slow phenomenon happening on the day to day timescale). This clarification was helpful.

They have added some (but not enough) direct support for the claim that each seizure is a phase transition. They have added nice examples showing how T (autocorrelation time) changes during single seizures from multiple patients (two patients in Fig 1 and many more in Supp Fig 2). I agree with the authors that these examples are consistent with the hypothesis that each seizure (at least at the onset) is a phase transition - marked by an initial increase in T and subsequent decrease in T as the seizure continues. However, these single examples do not constitute statistically rigorous support for the hypothesis. They are just 16 examples out of 2871 total seizures. Are these examples really representative of a statistically significant trend in the whole data set? This question must be resolved (with a YES answer) before I would consider the hypothesis well

supported.

Although there may be many ways to provide statistically sound support of the hypothesis, here is one possibility (I'm open to other ways). The authors could show the time course of T averaged over all seizure onsets, together with some representation of seizure-to-seizure variability. This might be nice to see on a grand average over all seizures, as well as for single patients. If these averages over many seizures do not show a significant rise and fall (or at least a significant abrupt fall like the green example in Fig 1) in T during seizure onset, this would cast doubt on the hypothesis that seizures are phase transitions.

Reviewers' comments:

Reviewer #1 (Remarks to the Author):

Authors responded to my comments and with the insertion of new definitions in the main text the readability of the draft has been increased.

I recommend the acceptance of the manuscript without further comments.

Reviewer #2 (Remarks to the Author):

This paper proposes to apply a canonical dynamic systems framework, of low-dimensional systems in the absence of noise, to intracranial EEG in patients with focal epilepsy. It explores the dynamics of a system close to a bifurcation that has a bistable region. A major limitation still remains in the manuscript. The study overlooks important aspects of the dynamics. Presumably, region s_3 (of Figure 1A) should exhibit oscillatory dynamics, which, in the absence of noise, would lead to a periodic autocorrelation. Then, Figure 1B becomes problematic because ACFW is not meaningful in this oscillatory regime (for s_3), and the diagram is misleading.

We thank the reviewer for this comment and the important concern raised which we fully agree with. However, the objective in this study was to monitor markers of critical slowing down that occur prior to a transition. To reiterate the goal of our study: Critical slowing down is predicted to occur under moderate noise, and when the approximation by linearization around the basin of attraction holds, as we state several times in our manuscript, e.g. in the introduction. Numerous models and studies have shown the validity of critical slowing for bistable systems and when it is expected to occur (for a comprehensive analytical treatise, please see e.g. Kuehn 2011 [6]). Critical slowing down is also predicted by numerous computational models of epileptic seizures, as discussed in our manuscript. Our central goal of our work was thus to monitor markers of critical slowing down in one of the most comprehensive human datasets and to compare these observations to predictions from critical transition theory.

Thus, the centrally relevant part of our claim is what is happening prior to the transition (i.e. leading up to points s_1 and s_2 in Figure 1). The predictions our model make are expected to occur regardless of what happens during the seizure in s_3 . Indeed, we observed critical slowing down in a statistical meaningful manner in 11/14 patients (supplementary figure 2) on short timescales, and also on a longer time scale in 9 of the 14 patients (Supplementary Figure 4). Our results thus show overwhelming support for critical slowing down and critical transitions at multiple time scales.

The reviewer states that the region S_3 exhibits oscillatory dynamics and therefore produces a meaningless autocorrelation. While the state S_3 is not centrally important to our claims (see above), we would nevertheless kindly like to reply that in many dynamical systems a transition

can occur in and out of an oscillatory regime where the autocorrelation function still maintains to contain important information about the dynamical state. For example:

Kuehn 2011 [6] provides a rigorous analytical treatise of several transitions (e.g. SNIC, subcritical Hopf) which clearly shows that critical slowing down is still expected to occur even when there is an oscillatory regime following. The results from the change in recovery rate for these different bifurcations obtained analytically also lead to a change in the autocorrelation function through the Wiener-Khinchin theorem. As a second example, Medeiros et al. 2017 (<https://www.nature.com/articles/srep42351.pdf>), analyzed phase transitions from bistable limit cycle systems. This work shows similarly how the autocorrelation function can be used in systems transitioning from one limit cycle to another limit cycle. In these systems, critical slowing down and a critical transition still occurs, but since the transition occurs from one limit cycle to another, the sharp drop at the transition point is not always observed. Finally, Meisel et al. 2015 [16] also demonstrated through modelling and experimental work that critical slowing down occurs can be observed, despite an oscillatory regime occurring after the transition. These are just some examples, there are many more to be found in the literature.

We acknowledge that the dynamics in s_3 can be complicated and in some cases might be oscillatory, in others it may be different. Our simple model (Figure 1), **assumed** that the states are changing from two stable non-oscillatory regimes (s_1 and s_3). The change in autocorrelation that we expect to observe given our assumption is explained in Figure 1. **Our assumption was then tested** in our work and shown to hold with significance for nearly all 2871 seizures (Supplementary Figure 2). A similar transition is also expected in a model like the epileptor (ref [23]), which is a bistable system with an oscillatory regime in the seizure state.

Given that we have not specified the dynamics in s_3 , we cannot make any claims about what occurs in s_3 . Therefore, we have softened the claim around what happens after the transition (lines 155-157):

...In many cases, the transition will be followed by a sharp drop in the width of the autocorrelation function assuming that the system transitions into a new state characterized by a faster system time constant.

We thank the reviewer for this comment and hope that he/she will appreciate the according changes made to the manuscript to make this aspect more clear.

Conceptually, noise is the key element to regulate transitions between stable branches. Its role must be very carefully considered.

The reviewer is correct that noise plays a key role. The careful consideration of noise is a central part of our work. First, the scaling laws of critical slowing down in terms of changes in autocorrelation and variance as the critical point is approached have been rigorously and analytically shown to hold with noise considered (in particular additive white noise; e.g. see Kuehn et al 2011; [6]). Thus, the observation of these changes in our empirical data is well in line with the theory of critical transitions. Second, noise is an important factor considered in our model and the interpretation of empirical data. Specifically, critical slowing down can only be observed when noise levels moderate (see Kuehn et al., 2011 for a rigorous analytical proof). The observation of critical slowing down in the majority of patients where we observe a

concomitant increase in ACFW and variance is thus in excellent agreement with these theoretical predictions under moderate noise. Third, we carefully consider the case of comparably high noise levels in our model in the transition from S1 direct to S3 (green curve in Figure 1A) and in our data. In the model, the dynamics “jumps” across the bifurcation point due to the large noise perturbations. Consequently, if noise (or perturbation) levels are too large, such as during the transition from S1 to S3 (green curve), no critical slowing down will be observed. In our data, this is likely to occur during the seizure transition in patients 5 and 7. (Note that throughout the text, we often use perturbations and noise synonymously, since both have the same effect on dynamics and it does not matter whether the noise source is intrinsic or extrinsic, i.e. by external perturbation. For a more analytical derivation of this equivalence, please see e.g. Kuehn et al., 2011. Therefore, large external perturbations can be equivalently seen in a dynamical sense as a form of noise.)

We agree with the reviewer that this dependence on noise should be made even more clear in the manuscript. We have thus added lines (136-139) to make this clearer and thank the reviewer for helping us to improve the manuscript.

In this case, we should observe critical slowing down - a slowing of the signals monitored, which is characterized by an increase in autocorrelation and variance (see Supplementary text for details). The occurrence of critical slowing down has been shown to occur under the assumption of moderate noise; noise that is too large can cause a transition to the new state [6]. The second involves a perturbation (e.g. noise) that kicks the system across the unstable threshold (dashed black line) and into the seizure state, s3 (green dashed arrow). In this case, a state transition still occurs but critical slowing down may not be expected due to the rapid push into a new state.

At the end, the differences between ACFW measured at s1 and s2 are very subtle, and the detection of transition to the other branch is trivial, and does not require any index.

This is an excellent point and was also brought up by reviewer 3. We believe the presentation of our results was not clear and it may have been difficult to appreciate the significant increase in critical slowing down markers prior to seizures due to the variability across patients. Based on the reviewer’s suggestions and comments, we have revisited the results to make these changes clearer and demonstrate statistical significance.

Specifically, on a short timescale, 11 of the 14 show a sharp increase in critical slowing down markers (ACFW, variance) prior to the seizure onset, thus providing strong evidence of critical slowing down prior to seizure onset. Note, that these changes were highly significant (Supplementary figure 2, patient average). Thus, while absolute changes might have appeared small due to the way they were presented in the previous version of the manuscript, these changes are definitely not subtle. Secondly, 13 of the 14 patients also showed a sharp drop of ACFW right after seizure onset, which is also expected from our model (Fig. 1) and can be seen as additional strong evidence for the occurrence of a critical transition at seizure onset.

Motivated by the excellent reviewer's comments, we have changed Supplementary Figure 2 to now show each patient's seizures (gray), and the average across all seizures (black). This should make the variability across seizures and patients more clear and demonstrate the large, significant changes in the vast majority of patients that is in excellent agreement with critical slowing down and seizure onset as a critical transition. We thank the reviewer for this comment which has helped us to make this important aspect of the manuscript more clear.

It seems that critical slowing down is relevant for continuous transitions, and not necessarily for bistable systems in the presence or absence of noise.

Numerous models and studies have shown the validity of critical slowing for bistable systems and when it is expected to occur (for a comprehensive analytical treatise, please see e.g. Kuehn 2011 [6]). Critical slowing down is also predicted by numerous computational models of epileptic seizures, as discussed in our manuscript. Our central goal of our work was thus to monitor markers of critical slowing down in one of the most comprehensive human datasets and to compare these observations to predictions from critical transition theory. Our results thus show overwhelming support for critical slowing down and critical transitions at multiple time scales.

We have demonstrated its relevance by (1) a statistical test showing that the changes in ACFW are significant (supplementary figure 2) and (2) using the markers of critical slowing down to produce the best prediction results yet obtained on this unique dataset, including Machine Learning algorithms. We hope the reviewer will appreciate these findings which we discuss in detail in the manuscript.

Reviewer #3 (Remarks to the Author):

The authors have substantially improved their paper. They effectively resolved all of my minor concerns. Regarding my Major Concern 1 (from first round of review), the authors have made good progress but have not yet sufficiently resolved this concern. To be clear, I will restate and slightly revise Major Concern 1:

Major Concern 1 (revised) A key part of the authors' work is their claim that they have new experimental support for the hypothesis that each seizure is a critical phase transition. In my view, this is the most interesting part of their findings. The additional work about long time dynamics and predicting seizures is also interesting, but it is not sufficiently ground-breaking to merit Nat Comms publication without the new support for the hypothesis that each seizure is a phase transition. Therefore, in my view, it is essential that the authors establish strong, statistically rigorous support for this hypothesis. In their revised manuscript the authors have made good progress towards this, but have not yet sufficiently done so.

They have clarified some previously-confusing concepts. It is now clear that they consider the phase transition to be a fast phenomenon that occurs for each seizure

onset (rather than a slow phenomenon happening on the day to day timescale). This clarification was helpful.

They have added some (but not enough) direct support for the claim that each seizure is a phase transition. They have added nice examples showing how T (autocorrelation time) changes during single seizures from multiple patients (two patients in Fig 1 and many more in Supp Fig 2). I agree with the authors that these examples are consistent with the hypothesis that each seizure (at least at the onset) is a phase transition - marked by an initial increase in T and subsequent decrease in T as the seizure continues. However, these single examples do not constitute statistically rigorous support for the hypothesis. They are just 16 examples out of 2871 total seizures. Are these examples really representative of a statistically significant trend in the whole data set? This question must be resolved (with a YES answer) before I would consider the hypothesis well supported.

We thank the reviewer for the overall positive assessment and our efforts to improve the manuscript. This reviewer raises an excellent point and, upon reviewing the presentation of our results in the previous manuscript version, we agree that these findings could have been presented in an even more clear way (please also see the related point made by reviewer 2). **The answer to the reviewer's question is YES.** Our supplementary Figure 2 actually showed all 2871 examples, not just 14 hand-picked examples. We agree that the figure text was perhaps unclear and have re-worded and changed the figure to make it clearer. (see below).

Although there may be many ways to provide statistically sound support of the hypothesis, here is one possibility (I'm open to other ways). The authors could show the time course of T averaged over all seizure onsets, together with some representation of seizure-to-seizure variability. This might be nice to see on a grand average over all seizures, as well as for single patients. If these averages over many seizures do not show a significant rise and fall (or at least a significant abrupt fall like the green example in Fig 1) in T during seizure onset, this would cast doubt on the hypothesis that seizures are phase transitions.

The reviewer has asked that we show *time course of T averaged over all seizure onsets*. We have changed Supplementary Figure 2 to show this. We feel this figure is now much clearer and clearly shows the *seizure-to-seizure variability*. The only case where we do not see convincing evidence of a phase transition is for patient 12.

We have added also a line in the main text to make it clearer that this figure shows **all seizures**:

Line 178: The presence of critical slowing down and critical transitions were analyzed for all seizures in each patient (Supplementary Figure 2).

We did further analysis to provide *statistically sound support* for seizures as a phase transition. This is shown in Supplementary Figure 2, where we report the patient average. The patient average was computed by first normalizing the ACFW by the average of a baseline period 5 minutes prior to the seizure. Then, we found the peak and trough during the seizure (red and

green dots Figure 1C, D and Figure 6C). The figure shows that the peak in ACFW at the seizure onset is significantly greater than baseline, and the drop in ACFW after the peak is significantly lower than baseline (Patient average in Supplementary Figure 2). We have explained this in the Results (lines 188-191) and Methods (lines 578-582) sections. The result shows that the peak in ACFW is significantly higher than the baseline period, and the subsequent drop is significantly lower than the baseline period.

As the reviewer may appreciate, the timescales over which critical slowing down occurs vary considerably between patients. As a result, a *grand average over all seizures* does not produce a meaningful result, as we hope the reviewer will agree. Instead we present the time course of critical slowing down markers for all seizures and all patients individually, exactly as the reviewer suggested. We thank the reviewer for this helpful comment which was important to improve our manuscripts and strengthen the claims therein.

As a side note, it may not have been clear that Supplementary Figure 4 also shows most seizures. In this figure, we focus on lead seizures only (definition explained in Methods) and show the average across all lead seizures (black) with standard error (gray shaded region). We have made the caption on this figure clearer.

REVIEWERS' COMMENTS:

Reviewer #2 (Remarks to the Author):

In this manuscript the authors study seizures using long-term intracranial electroencephalography recordings from patients with focal epilepsy. The four major findings are clearly indicated on page 20.

I cannot support publication of this manuscript in Nature Communications. There are many issues that I consider wrong or misleading in this manuscript. Here are the major ones:

Two of the four main findings (1 and 3) are not new. Empirical evidence for the role of variance, autocorrelation and correlation density exists for more than 10-20 years. See for example: Martinerie, Jacques, et al. "Epileptic seizures can be anticipated by non-linear analysis." *Nature medicine* 4.10 (1998): 1173-1176.

McSharry, Patrick E., Leonard A. Smith, and Lionel Tarassenko. "Prediction of epileptic seizures: are nonlinear methods relevant?." *Nature medicine* 9.3 (2003): 241-242.

Scheffer, Marten, et al. "Early-warning signals for critical transitions." *Nature* 461.7260 (2009): 53-59.

It is disappointing that these similar studies are not explored in the discussion.

The authors themselves have already shown similar results in a recent paper (figure 8):

Chang, Wei-Chih, et al. "Loss of neuronal network resilience precedes seizures and determines the ictogenic nature of interictal synaptic perturbations." *Nature neuroscience* 21.12 (2018): 1742-1752.

The fourth major finding seems promising, but the mechanism is not clear. An increase in overall brain excitability is too general, and the key questions are why and how does this clustering occur?

Reviewer #3 (Remarks to the Author):

The authors have done a nice job with their revisions and with clarifying some of my previous misunderstandings. I am now satisfied that they have shown convincing statistically significant support for the hypothesis that seizure onset is consistent with a phase transition. Considered together with the seizure prediction aspects of their work, my opinion is that this paper will make an impact and is ready to publish.

Reviewer #2 (Remarks to the Author):

In this manuscript the authors study seizures using long-term intracranial electroencephalography recordings from patients with focal epilepsy. The four major findings are clearly indicated on page 20.

I cannot support publication of this manuscript in Nature Communications. There are many issues that I consider wrong or misleading in this manuscript. Here are the major ones:

Two of the four main findings (1 and 3) are not new. Empirical evidence for the role of variance, autocorrelation and correlation density exists for more than 10-20 years. See for example:

Martinerie, Jacques, et al. "Epileptic seizures can be anticipated by non-linear analysis." *Nature medicine* 4.10 (1998): 1173-1176.

McSharry, Patrick E., Leonard A. Smith, and Lionel Tarassenko. "Prediction of epileptic seizures: are nonlinear methods relevant?." *Nature medicine* 9.3 (2003): 241-242.

Scheffer, Marten, et al. "Early-warning signals for critical transitions." *Nature* 461.7260 (2009): 53-59.

It is disappointing that these similar studies are not explored in the discussion.

The authors themselves have already shown similar results in a recent paper (figure 8): Chang, Wei-Chih, et al. "Loss of neuronal network resilience precedes seizures and determines the ictogenic nature of interictal synaptic perturbations." *Nature neuroscience* 21.12 (2018): 1742-1752.

The fourth major finding seems promising, but the mechanism is not clear. An increase in overall brain excitability is too general, and the key questions are why and how does this clustering occur?

We thank the reviewer for their comments over the course of article revisions. The comments have resulted in making our work much stronger. Please note that we had already included a discussion on the Scheffer 2009 paper. We have now also included the other two works by Martinerie and McSharry. Regarding why the clustering occurs. This is currently unknown but we are continuing to investigate. One possibility is that it is linked cycles in the immune response, which has similar duration cycles to the cycles observed in our study.

Reviewer #3 (Remarks to the Author):

The authors have done a nice job with their revisions and with clarifying some of my previous misunderstandings. I am now satisfied that they have shown convincing statistically significant support for the hypothesis that seizure onset is consistent with a phase transition. Considered together with the seizure prediction aspects of their work, my opinion is that this paper will make an impact and is ready to publish.

We thank the reviewer for their constructive comments over the course of article revisions. The comments have encouraged us to look at new aspects of critical slowing. As a result, we have shown overwhelming evidence that seizures conform to a critical phase transition.